

# Joint state-parameter estimation of a nonlinear stochastic energy balance model from sparse noisy data

Fei Lu[1], Nils Weitzel[2,3], and Adam H. Monahan[4]

[1]Department of Mathematics, Johns Hopkins University, Baltimore, Marlyand, USA
[2]Institut für Umweltphysik, Ruprecht-Karls-Universität Heidelberg, Heidelberg, Germany
[3]Institut für Geowissenschaften und Meteorologie, Rheinische Friedrich-Wilhelms-Universität Bonn, Bonn, Germany
[4]School of Earth and Ocean Sciences, University of Victoria, Victoria, British Columbia, Canada

**Correspondence:** Fei Lu (feilu@math.jhu.edu)

**Abstract.** While nonlinear stochastic partial differential equations arise naturally in spatiotemporal modeling, inference for such systems often faces two major challenges: sparse noisy data and ill-posedness of the inverse problem of parameter estimation. To overcome the challenges, we introduce a strongly regularized posterior by normalizing the likelihood and by imposing physical constraints through priors of the parameters and states.

5   We investigate joint parameter-state estimation by the regularized posterior in a physically motivated nonlinear stochastic energy balance model (SEBM) for paleoclimate reconstruction. The high-dimensional posterior is sampled by a particle Gibbs sampler that combines MCMC with an optimal particle filter exploiting the structure of the SEBM. In tests using either Gaussian or uniform priors based on the physical range of parameters, the regularized posteriors overcome the ill-posedness and lead to samples within physical ranges, quantifying the uncertainty in estimation. Due to the ill-posedness and the regularization, the posterior of parameters presents a relatively large uncertainty, and consequently, the maximum of the posterior, which is the minimizer in a variational approach, can have a large variation. In contrast, the posterior of states generally concentrates near the truth, substantially filtering out observation noise and reducing uncertainty in the unconstrained SEBM.

## 1   Introduction

Physically motivated nonlinear stochastic (partial) differential equations (SDEs and SPDEs) are natural models of spatiotem-
15  poral processes with uncertainty in geoscience. In particular, such models arise in the problem of reconstructing geophysical fields from sparse and noisy data (see e.g. Sigrist et al., 2015; Guillot et al., 2015; Tingley et al., 2012, and the references therein). The nonlinear differential equations, derived from physical principles, often come with unknown but physically con-strained parameters also to be determined from data. This promotes the problem of joint state-parameter estimation from sparse and noisy data. When the parameters are interrelated, which is often the case in nonlinear models, their estimation can be an
20  ill-posed inverse problem. Physical constraints on the parameters must then be taken into account. In variational approaches, physical constraints are imposed using a regularization term in a cost function, whose minimizer provides an estimator of the parameters and states. In a Bayesian approach, the physical constraints are encoded in prior distributions, extending the regularized cost function in the variational approach to a posterior and quantifying the estimation uncertainty. When the true parameters are known, the Bayesian approach has demonstrated great success in state estimation, thanks to the developments



in Monte Carlo sampling and data assimilation techniques (see e.g. Carrassi et al., 2018; Law et al., 2015; Vetra-Carvalho et al., 2018). However, the problem of joint state-parameter estimation, especially when the parameter estimation is ill-posed, has had relatively little success in nonlinear cases and remains a challenge (Kantas et al., 2009).

In this paper, we investigate a Bayesian approach for joint state and parameter estimation of a non-linear two-dimensional stochastic energy balance model (SEBM) in the context of spatial-temporal paleoclimate reconstructions of temperature fields from sparse and noisy data (Tingley and Huybers, 2010; Steiger et al., 2014; Fang and Li, 2016; Goosse et al., 2010). In particular, we consider a model of the energy balance of the atmosphere similar to those often used in idealized climate models (e.g. Fanning and Weaver, 1996; Weaver et al., 2001; Rypdal et al., 2015) to study climate variability and climate sensitivity. The use of such a model in paleoclimate reconstruction aims at improving the physical consistency of temperature
reconstructions during e.g. the last deglaciation and the Holocene by combining indirect observations, so called proxy data, with physically-motivated stochastic models.

    The SEBM models surface air temperature, explicitly taking into account sinks, sources, and horizontal transport of energy in the atmosphere, with an additive stochastic forcing incorporated to account for unresolved processes and scales. The model takes the form of a nonlinear SPDE with unknown parameters to be inferred from data. These unknown parameters are asso-
ciated with processes in the energy budget (e.g. radiative transfer, air-sea energy exchange) that are represented in a simplified manner in the SEBM, and may change with a changing climate. The parameters must fall in a prescribed range such that the SEBM is physically meaningful. Specifically, they must be in sufficiently close balance for the stationary temperature of the SEBM to be within a physically realistic range. As we will show, the parametric terms arising from this physically-based model are strongly correlated, leading to a Fisher information matrix that is ill-conditioned. Therefore, the parameter estimation is an
ill-posed inverse problem, and the maximum likelihood estimators of individual parameters have large variations and often fall out of the physical range.

    To overcome the ill-posedness in parameter estimation, we introduce a new strongly regularized posterior by normalizing the likelihood and by imposing the physical constraints through priors on the parameters and the states, based on physical constraints and the climatological distribution. In the regularized posterior, the prior has the same weight as the normalized
likelihood to enforce the support of the posterior to be in the physical range. Such a regularized posterior is a natural extension of the regularized cost function in a variational approach: the maximum of the posterior (MAP) is the same as the minimizer of the regularized cost function, but the posterior quantifies the uncertainty in the estimator.

    The regularized posterior of the states and parameters is high-dimensional and non-Gaussian. It is represented by its samples, which provide an empirical approximation of the distribution and allow efficient computation of quantities of interest such as
posterior means. The samples are drawn using a particle Gibbs sampler with ancestor sampling (PGAS, Lindsten et al., 2014), a special sampler in the family of particle Markov chain Monte Carlo (MCMC) methods (Andrieu et al., 2010) that combines the strengths of both MCMC and sequential Monte Carlo methods (see e.g. Doucet and Johansen, 2011) to ensure the convergence of the empirical approximation to the high-dimensional posterior. In the PGAS, we use an optimal particle filter that exploits the forward structure of the SEBM.



We consider two priors for the parameters, each based on their physical ranges: a uniform prior and a Gaussian prior with three standard deviations inside the range. We impose a prior for the states based on their overall climatological distribution. Tests show that the regularized posteriors overcome the ill-posedness and lead to samples of parameters and states within the physical ranges, quantifying the uncertainty in their estimation. Due to the regularization, the posterior of the parameters is

supported on a relatively large range. Consequently, the MAP of the parameters has a large variation, and it is important to use the posterior to assess the uncertainty. In contrast, the posterior of the states generally concentrates near the truth, substantially filtering out the observational noise and reducing the uncertainty in state reconstruction.

Tests also show that the regularized posterior is robust to spatial sparsity of observations, with sparser observations leading to larger uncertainties. However, due to the need for regularization to overcome ill-posedness, the uncertainty in the posterior

of the parameters can not be eliminated by increasing the number of observations in time. Therefore, we suggest alternative approaches, such as re-parametrization of the nonlinear function according to the climatological distribution or nonparametric Bayesian inference (see e.g. Müller and Mitra, 2013; Ghosal and Van der Vaart, 2017), to avoid ill-posedness.

The rest of the paper is organized as follows. Section 2 introduces the SEBM and its discretization, and formulates a state-space model. We also outline in this section the Bayesian approach to the joint parameter-state estimation and the particle

MCMC samplers. Section 3 analyzes the ill-posedness of the parameter estimation problem and introduces the regularized posterior. The regularized posterior is sampled by PGAS and numerical results are presented in Section 4. Discussions and conclusions are presented in Sections 5 and 6. Technical details of the estimation procedure are described in Appendix A.

## 2   State-space model formulation

After providing a brief physical introduction to the SEBM, we present its discretization and the observation model by repre-

senting them as a state-space model suitable for application of sequential Monte Carlo methods in Bayesian inference.

### 2.1   The stochastic energy balance model

The SEBM describes the evolution in space (both latitude and longitude) and time of the surface air temperature $u(t,\xi)$:

$$\partial_t u(t,\xi) - \nu \Delta u(t,\xi) = g_\theta(u) + f(t,\xi), \tag{1}$$

where $\xi \in [-\pi, \pi] \times [-\pi/2, \pi/2]$ is the two-dimensional coordinate on the sphere and the solution $u(t,\xi)$ is periodic in longi-

tude. Horizontal energy transport is represented as diffusion with diffusivity $\nu$, while sources and sinks of atmospheric internal energy are represented by the nonlinear function $g_\theta(u)$

$$g_\theta(u) = \theta_0 + \theta_1 u + \theta_4 u^4, \tag{2}$$

with the unknown parameters $\theta$. Upper and lower bounds of these three parameters, shown in Table 1, are derived from the energy balance model in Fanning and Weaver (1996), adjusted to current estimates of the Earth's global energy budget from

Trenberth et al. (2009) using appropriate simplifications. The equilibrium solution of the SEBM for the average values of the



**Table 1.** The physical upper and lower bounds of the parameters in the SEBM.

|             | $\theta_0$ | $\theta_1$ | $\theta_4$ |
|-------------|-------|--------|-------|
| upper bound | 32.57 | -22.70 | -4.80 |
| lower bound | 27.64 | -25.46 | -6.00 |

parameters approximates the current global mean temperature closely, and the magnitude of sinks and sources approximates the respective magnitudes in Trenberth et al. (2009) well. The physical ranges of the parameters are very conservative and cover current estimates of the global mean temperature during the Quaternary (Snyder, 2016). The state variable and the parameters in the model have been nondimensionalized so that the equilibrium solution of Eqn. (1) with $f = 0$ is approximately equal to
one.

The quartic nonlinearity of the function $g_\theta(u)$ arises from the Stefan-Boltzmann dependence of long-wave radiative fluxes on atmospheric temperature, while a linear feedback is included to represent state dependence of e.g. surface energy fluxes and albedo. Inclusion of quadratic and cubic nonlinarities in $g_\theta(u)$ (to account for nonlinearities in the feedbacks just noted) was found to exacerbate the ill-posedness of the model without qualitatively changing the character of the model dynamics within
the parameter range appropriate for the study of Quaternary climate variability (e.g. without admitting multiple deterministic equilibria associated with the ice-albedo feedback). In reality, the diffusivity $\nu$ and the parameters $\theta_j$, $j = (0, 1, 4)$ will depend on latitude, longitude, and time. We will neglect this complexity in our idealized analysis.

The stochastic term $f(t, \xi)$, which models the net effect of unresolved or oversimplified processes in the energy budget, is a centered Gaussian field that is white in time and colored in space, specified by an isotropic Matérn covariance function with
order $\alpha = 1$ and scale $\rho > 0$. That is,

$$\mathbb{E}\left[f(t, \xi)f(s, \eta)\right] = \delta(t - s)C(|\xi - \eta|), \tag{3}$$

with the covariance kernel $C(r)$ being the Matérn covariance kernel given by

$$C_\alpha(r) = \sigma_f^2 \frac{2^{1-\alpha}}{\Gamma(\alpha)} \left(\sqrt{2\alpha}\frac{r}{\rho}\right)^\alpha K_\alpha\left(\sqrt{2\alpha}\frac{r}{\rho}\right), \tag{4}$$

where $\Gamma$ is the gamma function, $\rho$ is a scaling factor, and $K_\alpha$ is the modified Bessel function of the second kind. We focus
on the estimation of the parameters $\theta$ and assume that $\nu$ and the parameters of $f$ are known. Estimating $\nu$ in energy balance models with data assimilation methods is studied in Annan et al. (2005), whereas estimation of parameters of $f$ in the context of linear SPDEs is covered for example in Lindgren et al. (2011).

In a paleoclimate context, temperature observations are sparse (in space and time) and derived from climatic proxies, such as pollen assemblages, isotopic compositions, and tree rings, that are indirect measures of the climate state. To simplify our
analysis, we neglect the potentially nonlinear transformations associated with the proxies and focus on the effect of observational sparseness. This is a common strategy in the testing of climate field reconstruction methods (e.g. Werner et al., 2013).





As such, we take the data to be noisy observations of the solution at $d_o$ locations:

$$y_i(t) = H_i(u(t)) + \epsilon_i(t) = u(t, \xi_i) + \epsilon_i(t), \tag{5}$$

for $i = 1, \ldots, d_o$, where each $\xi_i \in [-\pi, \pi] \times [-\pi/2, \pi/2]$ is a location of observation, $H$ is the observation operator, and $\epsilon_i(t) \sim \mathcal{N}(0, \sigma_\epsilon^2)$ are iid Gaussian noise. The data are sparse in the sense that only a small number of the spatial locations are observed.

## 2.2 State-space model representation

In practice, the differential equations are represented by their discretized systems and the observations are discrete in time, therefore we consider only the state space model based on a discretization of the SEBM. We refer the reader to Prakasa Rao (2001); Apte et al. (2007); Hairer et al. (2007); Maslowski and Tudor (2013); Llopis et al. (2018) for studies about inference of SPDEs in a continuous-time setting.

### 2.2.1 The state model

We discretize the SPDE (1) using linear finite elements in space and a semi-backward Euler method in time, using the computationally efficient Gaussian Markov random field approximation of the Gaussian field by Lindgren et al. (2011) (see details in Section A1). We write the discretized equation as a standard state space model:

$$U_{n+1} = \mu_\theta(U_n) + W_n \tag{6}$$

where $\mu_\theta : \mathbb{R}^{d_b} \to \mathbb{R}^{d_b}$ is the deterministic function and $\{W_n\}$ is a sequence of iid Gaussian noise with mean zero and covariance $\mathbf{R}$ described in more detail in Section (A19). Therefore, the transition probability density $p_\theta(u_{n+1}|u_n)$, the probability density of $U_{n+1}$ conditional on $U_n$ and $\theta$, is

$$p_\theta(u_{n+1}|u_n) = \det(2\pi\mathbf{R})^{-1/2} \exp\left(-\frac{(u_{n+1} - \mu_\theta(u_n))^T \mathbf{R}^{-1}(u_{n+1} - \mu_\theta(u_n))}{2}\right). \tag{7}$$

### 2.2.2 The observation model

In discrete form, we assume that the locations of observation are the nodes of the finite elements. Then the observation function in (5) is simply $H_i(U_n) = U_{n,k_i}$ with $k_i \in \{1, \ldots, d\}$ denoting the index of the node under observation, for $i = 1, \ldots, d_0$, and we can write the observation model as

$$Y_n = \mathbf{H}U_n + \epsilon_n, \quad y_n \in \mathbb{R}^{d_o}, \tag{8}$$

where $\mathbf{H} \in \mathbb{R}^{d_o \times d_b}$ is called the observation matrix, and $\{\epsilon_n\}$ is a sequence of iid Gaussian noise with distribution $\mathcal{N}(0, \mathbf{Q})$, where $\mathbf{Q} = \text{Diag}\{\sigma_i^2\}$. Equivalently, the probability of observing $y_n$ given state $U_n$ is

$$p(y_n|U_n) = \det(2\pi\mathbf{Q})^{-1/2} \exp\left(-\frac{(y_n - \mathbf{H}U_n)^T \mathbf{Q}^{-1}(y_n - \mathbf{H}U_n)}{2}\right). \tag{9}$$



## 2.3 Bayesian inference for SSM

Given observations $y_{1:N} := (y_1, \ldots, y_N)$, our goal is to jointly estimate the state $U_{1:N} := (U_1, \ldots, U_N)$ and the parameter vector $\theta := (\theta_0, \theta_1, \theta_4)$ in the state-space model (6)-(9). The Bayesian approach estimates the joint distribution of $(U_{1:N}, \theta)$ conditional on the observations by drawing samples to form an empirical approximation of the high-dimensional posterior. The empirical posterior efficiently quantifies the uncertainty in the estimation. Therefore, the Bayesian approach has been widely used (see the review Kantas et al., 2009, and the references therein).

Following Bayes' rule, the joint posterior distribution of $(U_{1:N}, \theta)$ can be written as

$$p(\theta, u_{1:N}|y_{1:N}) = p(\theta)\frac{p_\theta(u_{1:N})p_\theta(y_{1:N}|u_{1:N})}{p_\theta(y_{1:N})}, \tag{10}$$

where $p(\theta)$ is the prior of the parameters and $p_\theta(y_{1:N}) = \int p_\theta(u_{1:N})p_\theta(y_{1:N}|u_{1:N})du_{1:N}$ is the unknown marginal probability density function of the observations. In the importance sampling approximation to the posterior, we do not need to know the value of $p_\theta(y_{1:N})$, because as a normalizing constant, and it will be cancelled out in the importance weights of samples. The quantity $p_\theta(y_{1:N}|u_{1:N})$ is the likelihood of the observations $y_{1:N}$ conditional on the state $U_{1:N}$ and the parameter $\theta$, which can be explicitly derived from the observation model (8):

$$p_\theta(y_{1:N}|u_{1:N}) = p(y_{1:N}|u_{1:N}) = \prod_n p(y_n|u_n), \tag{11}$$

with $p(y_n|u_n)$ given in (9). Finally, the probability density function of the state $U_{1:N}$ given parameter $\theta$ can be derived from the state model (6):

$$p_\theta(u_{1:N}) = p_\theta(u_1) \prod_{n=1}^{N-1} p_\theta(u_{n+1}|u_n), \tag{12}$$

with $p_\theta(u_{n+1}|u_n)$ specified by (7).

## 2.4 Sampling the posterior by particle MCMC methods

In practice, we are interested in the expectation of quantities of interest or the probability of certain events. These computations involve integrations of the posterior that can neither be computed analytically nor by numerical quadrature methods due to the curse of dimensionality: the posterior is a high-dimensional non-Gaussian distribution involving variables with a dimension at the scale of thousands to millions. Monte Carlo methods generate samples to approximate the posterior by the empirical distribution, so that quantities of interest can be computed efficiently.

Markov Chain Monte Carlo (MCMC) methods are popular Monte Carlo methods (see e.g. Liu, 2001) that generate samples along a Markov chain with the posterior as the invariant measure. For joint distributions of parameters and states, a standard MCMC method is Gibbs sampling which consists of alternatively updating the state variable $U_{1:N}$ conditional on $\theta$ and $y_{1:N}$ by sampling

$$p(u_{1:N}|\theta, y_{1:N}) = \frac{p_\theta(u_{1:N})p_\theta(y_{1:N}|u_{1:N})}{p_\theta(y_{1:N})}, \tag{13}$$





and then updating the parameter $\theta$ conditional on $U_{1:N} = u_{1:N}$ by sampling the marginal posterior of $\theta$:

$$p(\theta|u_{1:N}, y_{1:N}) = p(\theta|u_{1:N}) = p(\theta)p_\theta(u_{1:N}). \qquad (14)$$

Due to the high-dimensionality of $U_{1:N}$, a major difficulty in sampling $p(u_{1:N}|\theta, y_{1:N})$ is the design of efficient proposal densities that can effectively explore the support of $p(u_{1:N}|\theta, y_{1:N})$.

Another group of rapidly-developing MC methods are sequential Monte Carlo (SMC) methods (Cappé et al., 2005; Doucet and Johansen, 2011) that exploit the sequential structure of state space models to approximate the posterior densities $p(u_{1:n}|\theta, y_{1:N})$ sequentially. SMC methods are efficient but suffer from the well-known problem of depletion (or degeneracy), in which the marginal distribution $p(u_n|\theta, y_{1:N})$ becomes concentrated on a single sample as $N - n$ increases (see Section A2 for more details).

The particle MCMC methods introduced in Andrieu et al. (2010) provide a framework for systematically combining SMC methods with MCMC methods, exploiting the strengths of both techniques. In the particle MCMC samplers, SMC algorithms provide high-dimensional proposal distributions, and Markov transitions guide the SMC ensemble to sufficiently explore the target distribution. The transition is realized by a conditional SMC technique, in which a reference trajectory from the previous step is kept throughout the current step of SMC sampling.

In this study, we sample the posterior by PGAS (Lindsten et al., 2014), a particle MCMC method that enhances the mixing of the Markov chain by sampling the ancestor of the reference trajectory. For the SMC, we use an optimal particle filter, which takes advantage of the linear Gaussian observation model and the Gaussian transition density of the state variables in our current SEBM. More generally, when the observation model is nonlinear and the transition density is non Gaussian, the optimal particle filter can be replaced by implicit particle filters (Chorin and Tu, 2009; Morzfeld et al., 2012) or local particle
filters (Penny and Miyoshi, 2016; Poterjoy, 2016); we refer to (Carrassi et al., 2018; Law et al., 2015; Vetra-Carvalho et al., 2018) for other data assimilation techniques. The details of the algorithm are provided in Section A3.

## 3   Ill-posedness and regularized posteriors

In this section, we first demonstrate and then analyze the failure of standard Bayesian inference of the parameters with the posteriors in (10). The standard Bayesian inference of the parameters fails in the sense that the posterior (10) tends to have
a large probability mass at non-physical parameter values. In the process of approximating the posterior by samples, the values of these samples often either hit the (upper or lower) bounds in Table 1 when we use a uniform prior or exceed these bounds when we use a Gaussian prior. As we shall show next, the standard Bayesian inverse problem is *numerically ill-posed* because the Fisher information matrix is ill-conditioned, which makes the inference numerically unreliable. Following the idea of regularization in variational approaches, we propose to use regularized posteriors in the Bayesian inference. This approach
unifies the Bayesian and the variational approaches: the MAP is the minimizer of the regularized cost function in the variational approach, but the Bayesian approach quantifies the uncertainty of the estimator by the posterior.





**Table 2.** The priors of $\theta = (\theta_0, \theta_1, \theta_4)$ based on the physical constraints in Table 1.

| Uniform prior | [27.64, 32.57]×[-25.46, -22.70]×[-6.00, -4.80] |
|---|---|
| Gaussian prior | mean $= (30.11, -24.08, -5.40)$ |
| | covariance $= \text{Diag}(0.82^2, 0.46^2, 0.20^2)$ |

**Table 3.** The settings of the stochastic energy balance model and its discretization.

| | | |
|---|---|---|
| $\nu$ | $= 0.1$ | Diffusion constant |
| $\sigma_f$ | $= 0.1$ | Scale of the stochastic forcing |
| $\Delta t$ | $= 0.01$ | Time step size |
| $d_b$ | $= 12$ | Number of total nodes |
| $d_o$ | $= 6$ | Number of observed nodes |
| $\sigma_\epsilon$ | $= 0.01$ | Std of the observation noise |

## 3.1 Model settings and tests

Based on the physical upper and lower bounds in Table 1, we consider two priors for the parameters: a uniform distribution on these intervals and a Gaussian distribution centered at the median and with three standard deviations in the interval, as listed in Table 2.

Throughout this study, we shall consider a relatively small numerical mesh for the SPDE with only 12 nodes for the finite elements. Such a small mesh provides a toy model that can neatly represent the spatial structure on the sphere, while allowing for systematic assessments of statistical properties of the Bayesian inference with moderate computational costs. Numerical tests show that the above FEM semi-backward Euler scheme is stable for a time step size $\Delta t = 0.01$ and a stochastic forcing with scale $\sigma_f = 0.1$ (see Section A1 for more details about the discretization). A typical realization of the solution is shown in

Figure 1 (left and middle), where we present the solution on the sphere at a fixed time with the 12-node finite element mesh, as well as the trajectories of all 12 nodes.

     The standard deviation of the observation noise is set to $\sigma_\epsilon = 0.01$, i.e. one order of magnitude smaller than the stochastic forcing and two orders of magnitude smaller than the climatological mean.

     We first assume that six out of the 12 nodes are observed; we discuss results obtained using sparser or denser observations

in the discussion section. Figure 1 also shows the climatological probability histogram of the true state variables and the partial noisy observations. The climatological distribution of the observations is close to that of the true state variables (with a slightly larger variance due to the noise). The histograms show that the state variables are centered around 1 and vary mostly in the interval $[0.92, 1.05]$. We shall use a Gaussian approximation based on the climatological distribution of the partial noisy observations as a prior to constrain the state variables.

We summarize the settings of numerical tests in Table 3.

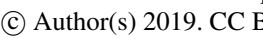


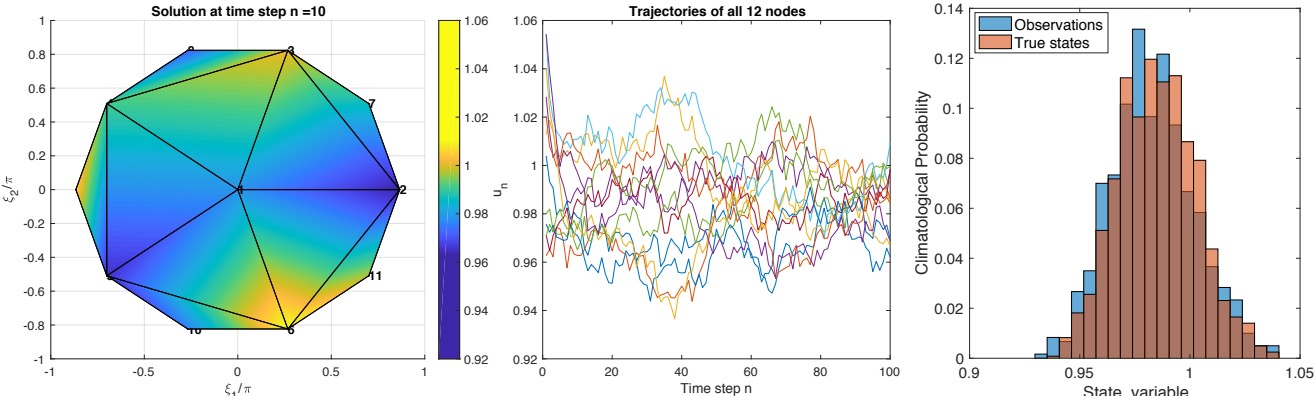

**Figure 1.** A typical realization of the solution to the SEBM. *Left:* the solution at time step $n = 10$ on the sphere with the 12-node finite element mesh. *Middle:* the trajectories of all 12 nodes over 100 time steps. *Right:* histogram estimates of the climatological probability distribution of all nodes of the true states (salmon) and the observations (blue).

## 3.2 Ill-posedness of the standard Bayesian inference of parameters

By the Bernstein-von Mises theorem (see e.g. Van der Vaart, 2000, Chaper 10), the posterior distribution of the parameters conditional on the true state data approaches the likelihood distribution as the data size increases. That is, $p(\theta|u_{1:N})$ in (14) becomes close to the likelihood distribution $p(u_{1:N}|\theta)$ (which can be viewed as a distribution of $\theta$) as the data size increases.

5    Therefore, if the likelihood distribution is numerically degenerate (in the sense that some components are undetermined), then the Bayesian posterior will also become close to degenerate, so that the Bayesian inference for parameter estimation will be ill-posed. In the following, we show that for this model the likelihood is degenerate even if the full states are observed with zero observation noise and that the maximum likelihood estimators have large nonphysical fluctuations (particularly when the states are noisy). As a consequence, the standard Bayesian parameter inference fails by yielding nonphysical samples.

10    We show first that the likelihood distribution is numerically degenerate because the Fisher information matrix is ill-conditioned. Following the transition density (7), the log-likelihood of the state $\{u_{1:N}\}$ is

$$l(\theta, u_{1:N}) = c - \frac{1}{2} \sum_{n=1}^{N} (u_{n+1} - \mu_\theta(u_n))^T \mathbf{R}^{-1} (u_{n+1} - \mu_\theta(u_n)), \tag{15}$$

where $c$ is a constant independent of $(\theta, u_{1:N})$. Since $\mu_\theta(\cdot)$ is linear in $\theta$ (cf. Equation (A19)), the likelihood function is quadratic in $\theta$ and the corresponding scaled Fisher information matrix is

$$\mathbf{F}_N = \frac{1}{N} \left( \sum_{n=1}^{N} G_{\theta,k}(u_n)^T \mathbf{R}^{-1} G_{\theta,l}(u_n) \right)_{k,l=0,1,4}, \tag{16}$$

where the vectors $G_{\theta,k}(u_n) \in \mathbb{R}^{d_b}$ are defined in (A20). Figure 2 shows the means and standard deviations of the condition numbers (the ratio between the maximum and the minimum singular values) of the Fisher information matrices from 100




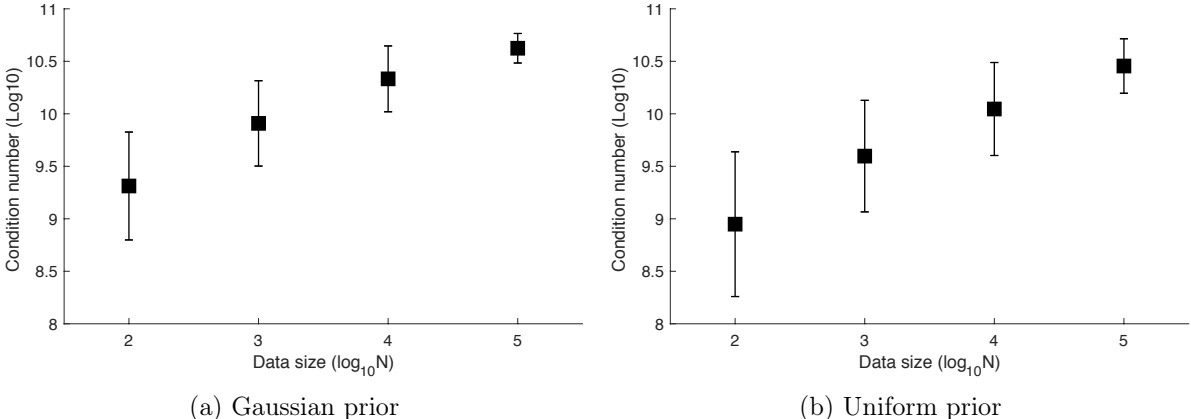

(a) Gaussian prior        (b) Uniform prior

**Figure 2.** The mean and standard deviation of the condition numbers of the Fisher information matrices, computed using true trajectories, out of 100 simulations of length ranging from $N = 10^2$ to $10^5$. The condition numbers are at the scale of $10^8 - 10^{11}$, indicating that the Fisher information matrix is ill-conditioned.

independent simulations. Each of these simulations generates a long trajectory of length $10^5$ using a parameter drawn randomly from the prior, and computes the Fisher information matrices using the true trajectory of all 12 nodes, for subsamples of lengths $N$ ranging from $10^2$ to $10^5$. For both Gaussian and uniform priors, the condition numbers are on the scale of $10^8 - 10^{11}$ and therefore the Fisher information matrix is ill-conditioned. In particular, the condition number increases as the data size

increased, due to the ill-posedness of the inverse problem of parameter estimation.

The ill-conditioned Fisher information matrix leads to highly variable maximum likelihood estimators (MLE), computed from $\mathbf{F}_N \theta = b_N$ with $b_N = \frac{1}{N} \left( \sum_{n=1}^{N} G_{\theta,k}(u_n)^T \mathbf{R}^{-1}(u_{n+1} - \mathbf{M}_{\Delta t}^{-1}\mathbf{M}_0 u_n) \right)_{k=0,1,4}$, which follows from (A20).

The ill-posedness is particularly problematic when $\{u_{1:N}\}$ is observed with noise, as the ill-conditioned Fisher information matrix amplifies the noise in observations and leads to nonphysical estimators. Figure 3 shows the means and standard devia-

tions of errors of MLEs computed from true and noisy trajectories in 100 independent simulations. In each of these simulations, the "noisy" trajectory is obtained by adding a white noise with standard deviation $\sigma_\epsilon = 0.01$ to a "true" trajectory generated from the system with a true parameter randomly drawn from the prior. For both Gaussian and uniform priors, the standard deviations and means of the errors of the MLE from the noisy trajectories are one order of magnitude larger than those from true trajectories. In particular, the variations are large when the data size is small. For example, when $N = 100$, the standard

deviation of the MLE for $\theta_0$ from noisy observations is on the order of $10^3$, two orders of magnitude larger than its physical range in Table 2. The standard deviations decrease as the data size increases, at the expected rate of $1/\sqrt{N}$. However, the errors are too large to be practically reduced by increasing the size of data: for example, a data size $N = 10^{10}$ is needed to reduce the standard deviation of $\theta_4$ to less than $0.1$ (which is about 10% the size of the physical range $[-6.00, -4.80]$ as specified in Table 2). In summary, the ill-posedness leads to parameter estimators with large variations that are far outside the physical ranges of

the parameters.


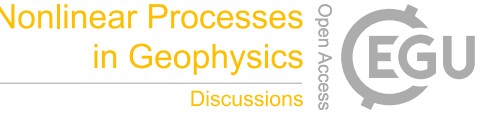

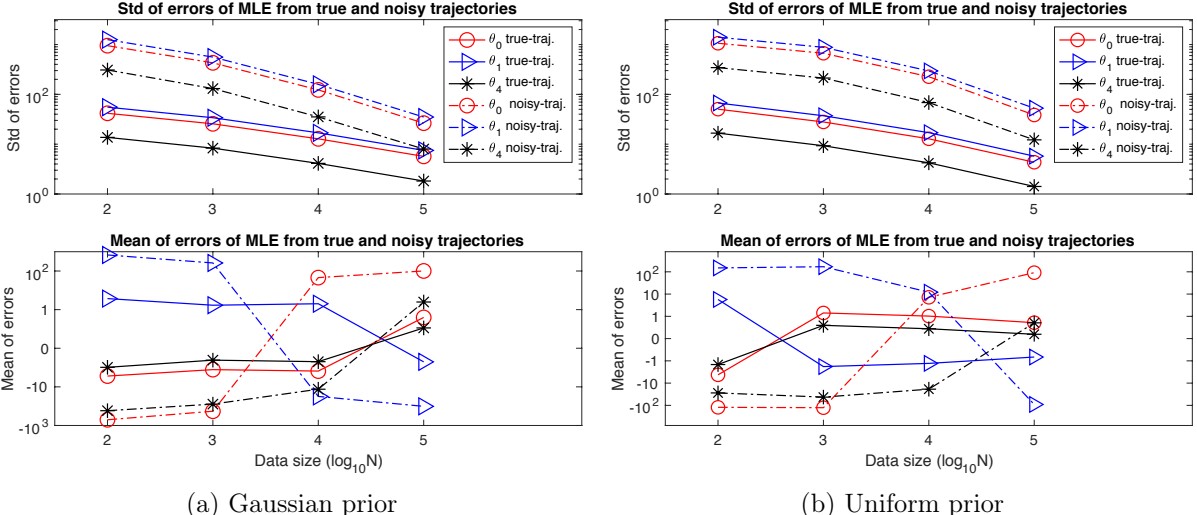

(a) Gaussian prior                    (b) Uniform prior

**Figure 3.** The standard deviations and means of the errors of the MLEs, computed from true and noisy trajectories, out of 100 independent simulations with true parameters sampled from the Gaussian and uniform priors. In all cases, the deviations and biases (i.e. means of errors) are large. In particular, in the case of noisy observations, the deviations are at orders ranging from 10 to 1000, far beyond the physical ranges of the parameters in Table 1. Though the deviations decrease as data size increases, an impractically large data size is needed to reduce them to a physical range. Also, the means of errors are larger than the size of physical ranges of the parameters with values that decay slowly as data size increases.

### 3.3 Regularized posteriors

To overcome the ill-posedness of the parameter estimation problem, we introduce strongly regularized posteriors by normalizing the likelihood function. In addition, to prevent unphysical values of the states, we further regularize the state variables in the likelihood by an uninformative climatological prior. That is, consider the *regularized posterior*:

$$p^N(\theta, u_{1:N}|y_{1:N}) = \frac{1}{Z}p(\theta)\left[\frac{p^c(u_{1:N})p_\theta(u_{1:N})p_\theta(y_{1:N}|u_{1:N})}{p_\theta(y_{1:N})}\right]^{1/N}, \tag{17}$$

where $Z := \int p(\theta)\left[\frac{p^c(u_{1:N})p_\theta(u_{1:N})p_\theta(y_{1:N}|u_{1:N})}{p_\theta(y_{1:N})}\right]^{1/N} d\theta du_{1:N}$ is a normalizing constant and $p^c(u_{1:N})$ is the prior of the states estimated from a Gaussian fit to climatological statistics of the observations, neglecting correlations. That is, we set $p^c(u_{1:N})$ as

$$p^c(u_{1:N}) := \prod_{i=1}^{N}\frac{1}{2\pi\sigma_c^{d_b}}\exp\left(-\frac{|u_i - u_c|^2}{2\sigma_c^2}\right) \tag{18}$$

10    with $\sigma_c = 2\sqrt{\sigma_o^2 - \sigma_\epsilon^2}$, where $u_c$ and $\sigma_o$ are the mean and standard deviation of the observations over all states. Here the multiplicative factor 2 aims for a larger band to avoid an overly narrow prior for the states.

This prior can be viewed as a joint distribution of the state variables assuming all components are independent identically Gaussian distributed with mean $u_c$ and variance $\sigma_c^2$. Clearly, it uses the minimum amount of information about the state



variables, and we expect it can be improved by taking into consideration spatial correlations or additional field knowledge in practice.

The regularized posterior can be viewed as an extension of the regularized cost function in the variational approach. In fact, the negative logarithm of the regularized posterior is the same (up to a multiplicative factor $\frac{1}{N}$ and an additive constant

$\log Z - \frac{1}{N} \log p_\theta(y_{1:N})$) as the cost function in variational approaches with regularization. More precisely, we have

$$-\log p^N(\theta, u_{1:N}|y_{1:N}) \;=\; \frac{1}{N} C_{y_{1:N}}(\theta, u_{1:N}) + \log Z - \frac{1}{N}\log p_\theta(y_{1:N}), \tag{19}$$

where $C_{y_{1:N}}(\theta, u_{1:N})$ is the cost function with regularization:

$$C_{y_{1:N}}(\theta, u_{1:N}) \;=\; -\sum_{n=1}^{N} \log\big[p(u_n|u_{n-1}, \theta)p(y_n|u_n)\big] - N\log p(\theta) - \log p^c(u_{1:N}). \tag{20}$$

When the prior is Gaussian, the regularization corresponds to Tikhonov regularization. Therefore, the regularized posterior ex-

tends the regularized cost function to a probability distribution, with the maximum of the posterior (MAP) being the minimizer of the regularized cost function.

The regularized posterior normalizes the likelihood by an exponent $1/N$. This normalization allows for a larger weight (more trust) on the prior, which can then sufficiently regularize the singularity in the likelihood and therefore reduces the probability of nonphysical samples. Intuitively, it avoids the shrinking of the likelihood as the data size increases. When the system is

ergodic, the sum $\frac{1}{N}\sum_{n=1}^{N}\log\big[p_\theta(u_n|u_{n-1})p(y_n|u_n)\big]$ converges to the spatial average $\mathbb{E}[\log\big[p_\theta(U_n|U_{n-1})p(y_n|U_n)\big]$ with respect to the invariant measure as $N$ increases. While being effective, this factor may not be optimal (O'Leary, 2001) and we leave the exploration of optimal regularization factors to future work.

In the sampling of the regularized posterior, we update the state variable $U_{1:N}$ conditionally on $\theta$ and $y_{1:N}$ by sampling $p^c(u_{1:N})p_\theta(u_{1:N}|\theta, y_{1:N})$ (with $p_\theta(u_{1:N}|\theta, y_{1:N})$ specified in (13)) using SMC methods. Compared to the standard PMCMC

algorithm outlined in Section 2.4, the only difference occurs when we update the parameter $\theta$ conditional on the estimated states $u_{1:N}$. Instead of (14), we draw a sample of $\theta$ from the regularized posterior

$$p^N(\theta|u_{1:N}, y_{1:N}) \;\propto\; p(\theta)[p_\theta(u_{1:N})]^{1/N}. \tag{21}$$

## 4 Bayesian inference with regularized posteriors

The regularized posteriors are approximated by the empirical distribution of samples drawn using particle MCMC methods,

specifically particle Gibbs with ancestor sampling (PGAS, see Section A3) in combination with SMC using optimal importance sampling (see Section A2). In the following section, we first diagnose the Markov chain and choose a reasonable chain length for subsequent analyses. We then present the results of parameter estimation and state estimation.

In all the tests presented in this study, we use only $M = 5$ particles for the SMC, as we can be confident of the Markov chain produced by the particle MCMC methods converging to the target distribution based on theoretical results (see Andrieu



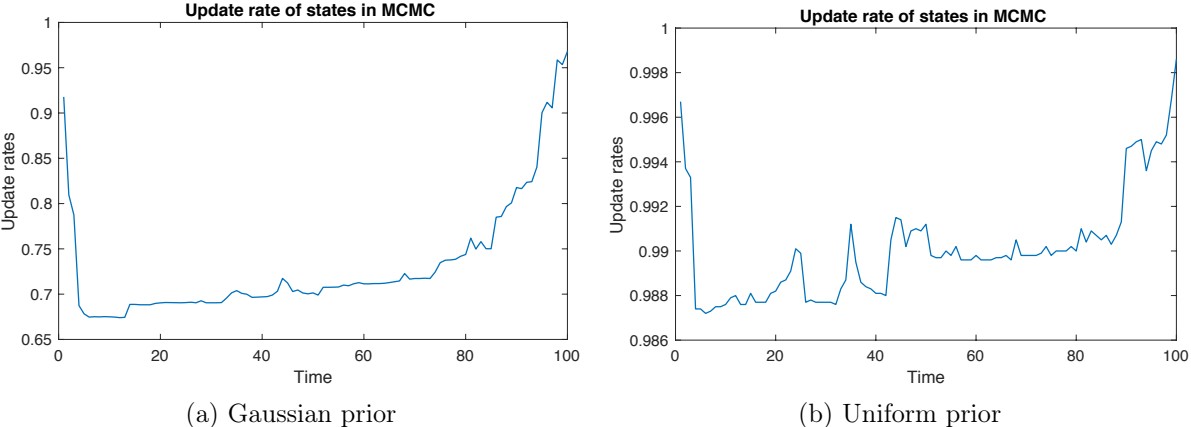

**Figure 4.** The update rate of the states at different times along the trajectory. The high update rate at time $t = 1$ is due to the initialization of the particles near the equilibrium and the ancestor sampling. The high update rate at the end time is due to the nature of the SMC filter. Note that the uniform prior has update rates close to 1 at all times.

et al., 2010; Lindsten et al., 2014). In general, the more particles are used, the better the SMC algorithm (and hence the particle MCMC methods) will perform, at the price of increased computational cost.

## 4.1 Diagnosis of the Markov Chain

To ensure that the Markov Chain generated by PGAS is well-mixed and to find a length for the chain such that the posterior
is acceptably approximated, we shall assess the Markov chain by three criteria: the update rate of states; the correlation length of the Markov chain; and the convergence of the marginal posteriors of the parameters. These empirical criteria are convenient and, as we discuss below, have found to be effective in our study. We refer to Cowles and Carlin (1996) for a detailed review of various criteria for diagnosing MCMC.

    The update rate of states is computed at each time of the state trajectory $u_{1:N}$ along the Markov chain. That is, at each
time, we say the state is updated from the previous step of of the Markov chain if any entry of the state vector changes. The update rate measures the mixing of the Markov chain. In general, an update rate above 0.5 is preferred, but a high rate close to 1 is not necessarily the best. Figure 4 shows the update rates of typical simulations for both the Gaussian prior and the uniform prior. For both priors, the update rates are above 0.5, indicating a fast mixing of the chain. The rates tend to increase with time (except for the first time step) to a value close to 1 at the end of the trajectory. This phenomenon agrees with the
particle depletion nature of the SMC filter: when tracing back in time to sample the ancestors, there are fewer particles and therefore the update rate is lower. The high update rate at time $t = 1$ step is due to our initialization of the particles near the equilibrium, which increases the possibility of ancestor updates in PGAS. We also note that the uniform prior has update rates close to 1 at all times, much higher than the rates of the Gaussian prior. Higher update rates occur for the uniform prior because

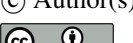



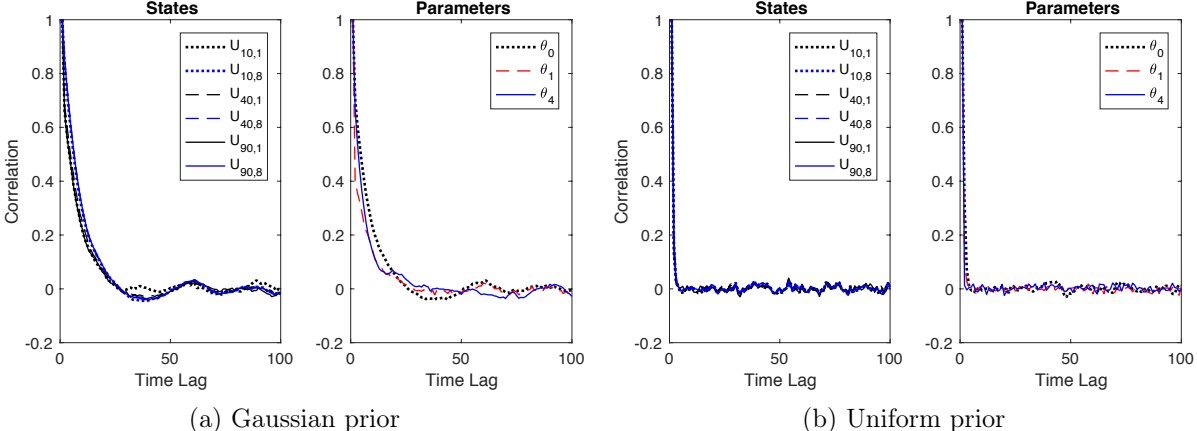

**Figure 5.** The empirical autocorrelation functions (ACF) of the Markov chain of parameters $(\theta_0, \theta_1, \theta_4)$ and states $U_{n,k}$ at times $n = \{10, 40, 90\}$ and notes $k = \{1, 8\}$, computed from a Markov chain with length 10000. The ACFs fall within a threshold of 0.1 around zero within a time lag about 25 for the Gaussian prior, and a time lag about 5 for the uniform prior.

**Table 4.** The settings of the particle MCMC using SMC with optimal importance densities.

| $M$ | $= 5$ | Number particles in SMC |
|---|---|---|
| $L$ | $= 10^4$ | Length of the Markov chain |
| $N$ | $= 100$ | Number of time steps of observations. |

the deviations of parameter samples from the previous values are larger, resulting in an increased probability of updating the reference trajectory in the conditional SMC.

We test the correlation length of the Markov chain by finding the smallest lag at which the empirical autocorrelation functions (ACF) of the states and the parameters are close to zero. Figure 5 shows the empirical ACFs of the parameters and states at

5   representative nodes, computed using a Markov chain with length 10000. The ACFs approach zero within a time lag of around 40 ( based on a threshold value of 0.1) for the Gaussian prior, and within a time lag of around 5 for the uniform prior. The smaller correlation length in the uniform prior case is again due to the larger parameter variation in the uniform prior case than the Gaussian prior case.

The relatively small decorrelation length of the Markov chain indicates that we can accurately approximate the posterior by

10   a chain of a relatively short length. This result is demonstrated in Figure 6, where we plot the empirical marginal posteriors of the parameters, using Markov chains of three different lengths: $L = 1000, 5000, 10000$. The marginal posteriors with $L = 1000$ are reasonably close to those with $L = 10^4$, and those with $L = 5000$ are almost identical to those with $L = 10^4$. In particular the marginal posteriors with $L = 10^3$ capture the shape and spread of the distributions for $L = 10^4$. Therefore, a Markov chain with length $L = 10^4$ provides a reasonably accurate approximation of the posterior. Hence, we use Markov chains with length





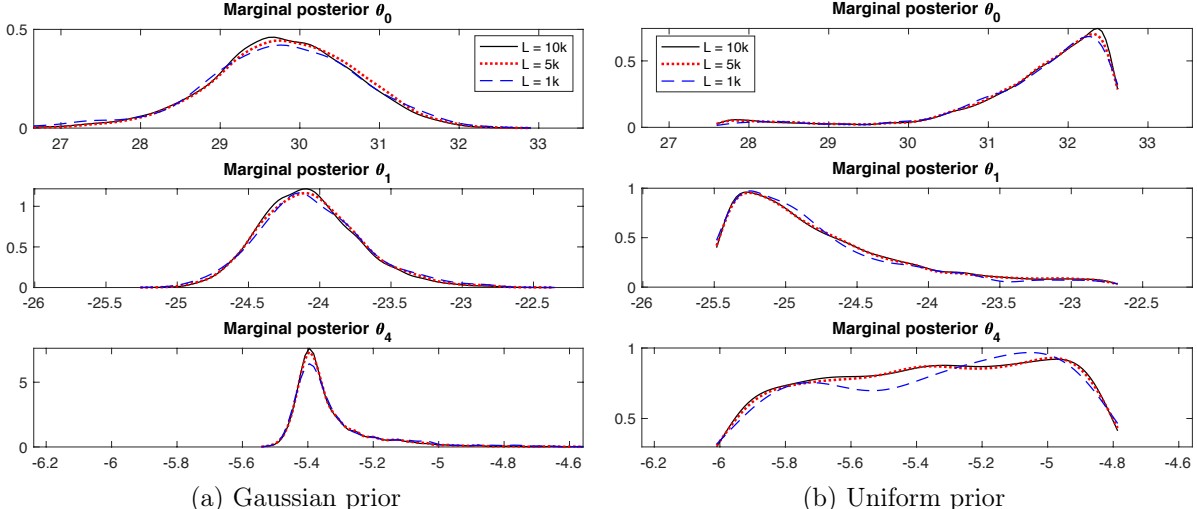

**Figure 6.** The empirical marginal distributions of the samples from the posterior as the length of the Markov chain increases. Note that the marginal posteriors converge rapidly as the length of the chain increases. In particular, a chain with length 1000 provides a reasonable approximation to the posterior, capturing the shape and spread of the distribution.

$L = 10^4$ in all simulations from here on. This choice of chain length may be longer than necessary, but allows for confidence that the results are robust.

In summary, based on the above diagnosis of the Markov chain generated by PMCMC, to run many simulations for statistical analysis of the algorithm within a limited computation cost, we use chains with length $L = 10^4$ to approximate the posterior.

For the SMC algorithm, we use only five particles. The number of observations in time is $N = 100$.

## 4.2  Parameter estimation

One of the main goals in Bayesian inference is to quantify the uncertainty in the parameter-state estimation by the posterior. We access the parameter estimation by examining the samples of the posterior in a typical simulation, for which we consider the scatter plots and marginal distributions, the maximum of the posterior (MAP) and the posterior mean. We also examine

the statistics of the MAP and the posterior mean in 100 independent simulations. In each simulation, the parameters are drawn from the prior distribution of $\theta$. Then, a realization of the SEBM is simulated. Finally, observations are created by applying the observation model to the SEBM realization.

The empirical marginal posteriors of the parameters $\theta = (\theta_0, \theta_1, \theta_4)$ in two typical simulations, for the Gaussian and the uniform priors, are shown in Figure 7. The top row presents scatter plots of samples along with the true values of the parameters

(asterisks), and the bottom row presents the marginal posteriors for each parameter in comparison with the priors.

In the case of the Gaussian prior, the scatter plots show a posterior that is far from Gaussian, with clear nonlinear dependence between $\theta_0$ and the other parameters. The marginal posteriors of $\theta_0$ and $\theta_1$ are close to their priors, with larger tails (to the left





(a) Gaussian prior  (b) Uniform prior

**Figure 7.** Posteriors of the parameters in a typical simulation, with both the Gaussian and the uniform prior. The true values of the parameters, as well as the data trajectory, are the same for both priors. The top row displays scatter plots of the samples (blue dots), with the true values of the parameters shown by asterisks. The bottom row displays the marginal posteriors (blue lines) of each component of the parameters and the priors (black dash-dot lines), with the posterior mean marked by diamonds and the true values marked by asterisks. The posterior correlations are $\rho_{01} = 0.20$, $\rho_{04} = -0.19$ and $\rho_{14} = 0.57$ in the case of Gaussian prior; and $\rho_{01} = -0.23$, $\rho_{04} = -0.01$ and $\rho_{14} = -0.05$ in the case of uniform prior.

for $\theta_0$ and to the right for $\theta_1$). The marginal distribution of $\theta_4$ concentrates near the center of the prior with larger tail to the right. The posterior has the most probability mass near the true values of $\theta_0$ and $\theta_1$, which are in the high probability region of the prior. However, it has no probability mass near the true value of $\theta_4$ – which is of a low probability in the prior.

In the case of the uniform prior, the scatter plots show a concentration of probability near the boundaries of the physical range. The marginal posteriors of $\theta_0$ and $\theta_1$ clearly deviate from the priors, concentrating near the parameter bounds (the upper



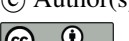

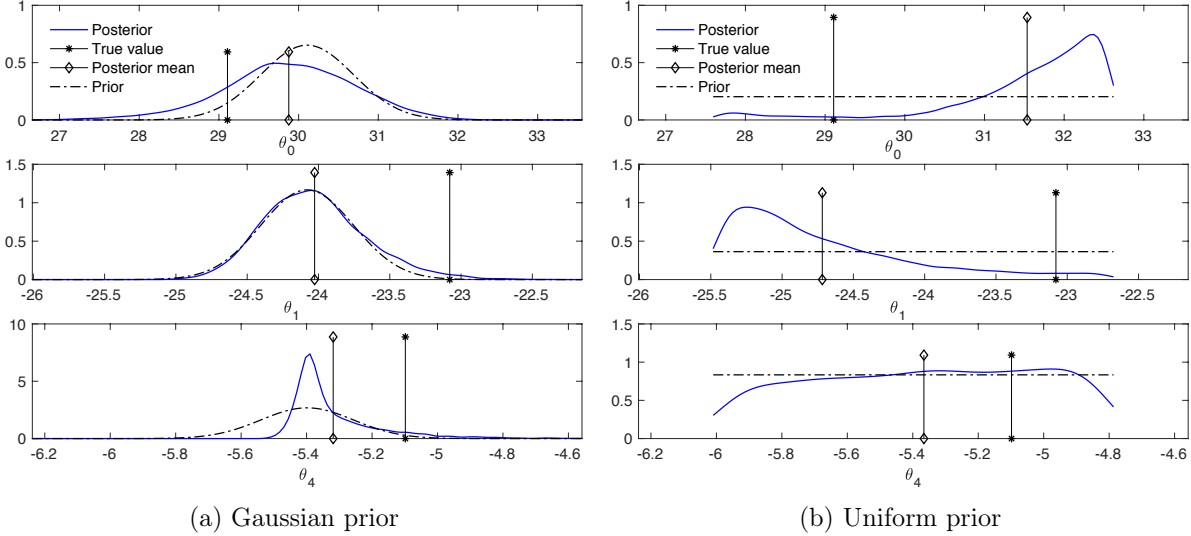

(a) Gaussian prior           (b) Uniform prior

**Figure 8.** The marginal posteriors with a different set of true values for the parameters. The marginal posteriors change little from those in Fig. 7.

bound for $\theta_0$ and the lower bound for $\theta_1$ in this realization); the marginal posterior of $\theta_4$ is close to the prior with slightly more probability mass for large values.

Further tests show that the posterior is not sensitive to changes in the true values of the parameters. This fact is demonstrated in Figure 8, which presents the marginal distributions for another set of true values of the parameters (but without changing the priors). Though the data change when the true parameters change, the posteriors, in comparison with those Figure 7, change little for both cases of Gaussian and uniform prior.

The non-Gaussianity of the posterior (including the concentration near the boundaries), its insensitivity to changes in the true parameter, and its limited reduction of uncertainty from the prior (Figures 7 - 8) are due to the degeneracy of the likelihood distribution and to the strong regularization. Recall that the degenerate likelihood leads to MLEs with large variations and biases, with the standard deviation of the estimators of $\theta_0$ and $\theta_1$ being about 10 times larger than those of $\theta_4$ (see Figure 3). As a result, when regularized by the Gaussian prior, the components $\theta_0$ and $\theta_1$, which are more under-determined by the likelihood, are constrained mainly by the Gaussian prior and therefore their marginal posteriors are close to their marginal priors. In contrast, the component $\theta_4$ is forced to concentrate around the center of the prior but with a large tail. While dramatically reducing the large uncertainty of $\theta_0$ and $\theta_1$ in the ill-conditioned likelihood, the regularized posterior still exhibits a slightly larger uncertainty than the prior for these two components.

In the case of the uniform prior, it is particularly noteworthy that the marginal posteriors of $\theta_0$ and $\theta_1$ differ more from their priors than the parameter $\theta_4$. These results are the opposite of what was found for the Gaussian prior. Such differences are due to the different mechanism of "regularization" by the two priors. The Gaussian prior eliminates the ill-posedness by





**Table 5.** Means and standard deviations of the errors of the posterior mean and MAP in 100 independent simulations.

**(a) The case of observing six of the 12 nodes.**

|  |  | $\theta_0$ | $\theta_1$ | $\theta_4$ |
|---|---|---|---|---|
| Gauss Prior | Posterior mean | -0.44 ± 0.58 | 0.09 ± 0.42 | 0.11 ± 0.20 |
|  | MAP | -0.32 ± 0.61 | 0.02 ± 0.42 | 0.03 ± 0.21 |
| Uniform Prior | Posterior mean | 0.75 ± 1.06 | -0.31 ± 1.07 | -0.02 ± 0.35 |
|  | MAP | 1.02 ± 1.53 | -0.51 ± 1.49 | 0.15 ± 0.43 |

**(b) The case of observing two of the 12 nodes.**

|  |  | $\theta_0$ | $\theta_1$ | $\theta_4$ |
|---|---|---|---|---|
| Gauss Prior | Posterior mean | -0.32 ± 0.61 | -0.03 ± 0.37 | 0.10 ± 0.20 |
|  | MAP | -0.19 ± 0.67 | -0.10 ± 0.38 | 0.02 ± 0.20 |
| Uniform Prior | Posterior mean | 0.77 ± 1.12 | -0.39 ± 1.00 | 0.07 ± 0.36 |
|  | MAP | 1.06 ± 1.55 | -0.61 ± 1.42 | 0.27 ± 0.42 |

regularizing the ill-conditioned Fisher information matrix with the covariance of the prior. So, the information in the likelihood, e.g. the bias and the correlations between $(\theta_0, \theta_1)$ and $\theta_4$, are preserved in the regularized posterior. The uniform prior, on the other hand, cuts the support of the degenerate likelihood and rejects out-of-range samples. As a result, the correlation between $\theta_0$ and $\theta_1$ is preserved in the regularized posterior because they feature similar variations, but the correlations between $(\theta_0, \theta_1)$

and $\theta_4$ are weakened (Figure 7).

    In practice, one is often interested in a point estimate of parameters. Commonly used point estimators are the MAP and the posterior mean. Figures 7-8 show that both the MAP and the posterior mean can be far away from the truth for Gaussian as well as uniform priors. In particular, in the case of the uniform prior, the MAP values are further away from the truth than the posterior mean. In the case of the Gaussian prior, the MAP values do not present a clear advantage or disadvantage over the

posterior mean.

    Table 5(a) shows the means and standard deviations of the errors of the posterior mean and MAP from 100 independent simulations. In each simulation and for each prior, we drew a parameter sample from the prior and generated a trajectory of observations, and then estimated jointly the parameters and states. The table shows that both posterior mean and MAP estimates are generally biased, consistent with the biases in Figures 7 and 8. More specifically, in the case of the Gaussian prior, the MAP

has slightly smaller biases than the posterior mean, but the two have almost the same variances. Both are negatively biased for $\theta_0$ and slightly positively biased for $\theta_1$ and $\theta_4$. In the case of the uniform prior, the MAP features biases and standard deviations which are about 50% larger than those of the posterior mean. Both estimators exhibit large positive biases in $\theta_0$, large negative biases in $\theta_1$, and small positive biases in $\theta_4$.



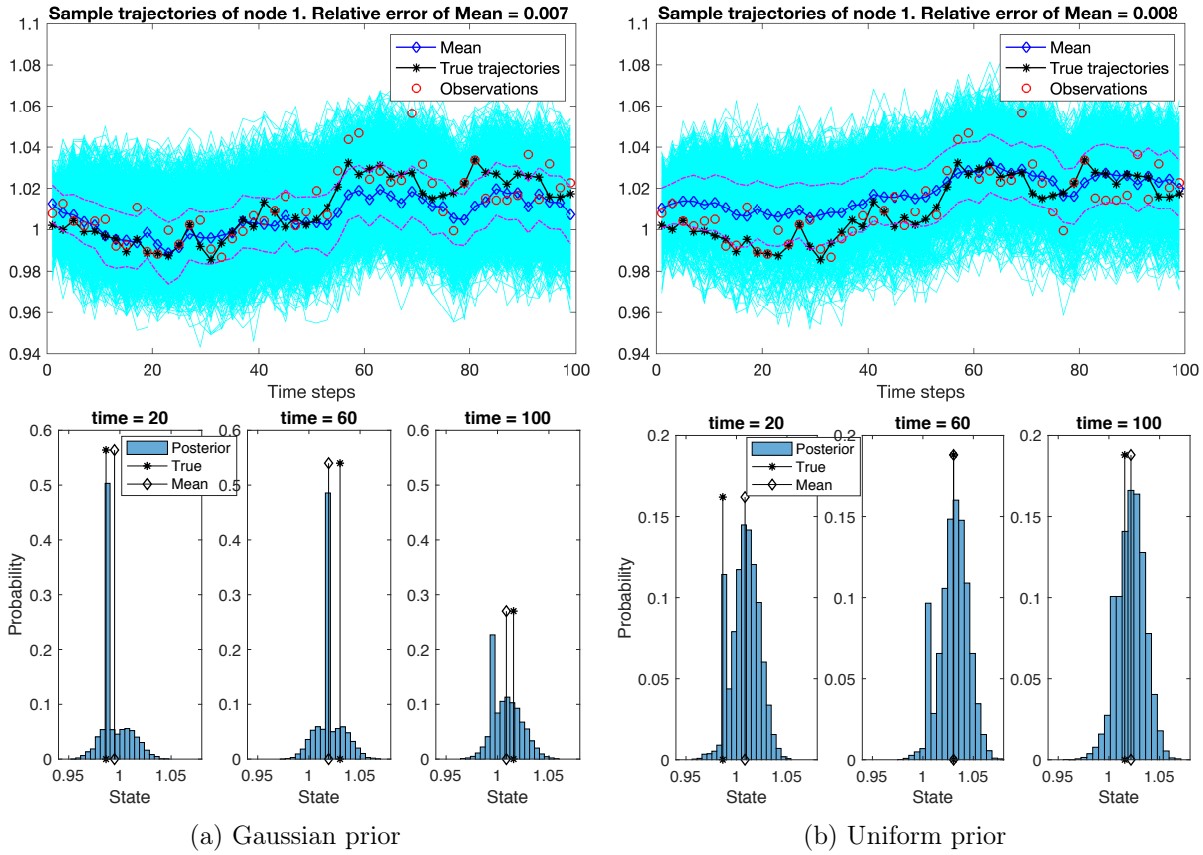

(a) Gaussian prior                               (b) Uniform prior

**Figure 9.** The ensemble of sample trajectories of the state at an observed node. Top row: the sample trajectories (in cyan) concentrate around the true trajectory (in black dash-asterisk). The true trajectory is well-estimated by the ensemble mean (in blue dash-diamond), and is mostly enclosed by the one-standard-deviation band (in magenta dash-dot lines). The relative error of the ensemble mean along the trajectory is 0.7% and 0.8%, filtering out 30% and 20% of the observation noise, respectively. Bottom row: histograms of samples at three instants of time: $t = 20$, $t = 60$ and $t = 100$. The histograms show that the samples concentrate around the true states.

## 4.3 State estimates

The state estimation aims both to filter out the noise from the observed nodes and to estimate the states of unobserved nodes. We access the state estimation by examining the ensemble of the posterior trajectories in a typical simulation, for which we consider the marginal distributions and the coverage probability of 90% credible intervals. We also examine the statistics of

5  these quantities in 100 independent simulations.

We present the ensemble of posterior trajectories at an observed node in Figure 9 and at an unobserved node in Figure 10. In each of these figures, we present the ensemble mean with a one-standard-deviation band, in comparison with the true





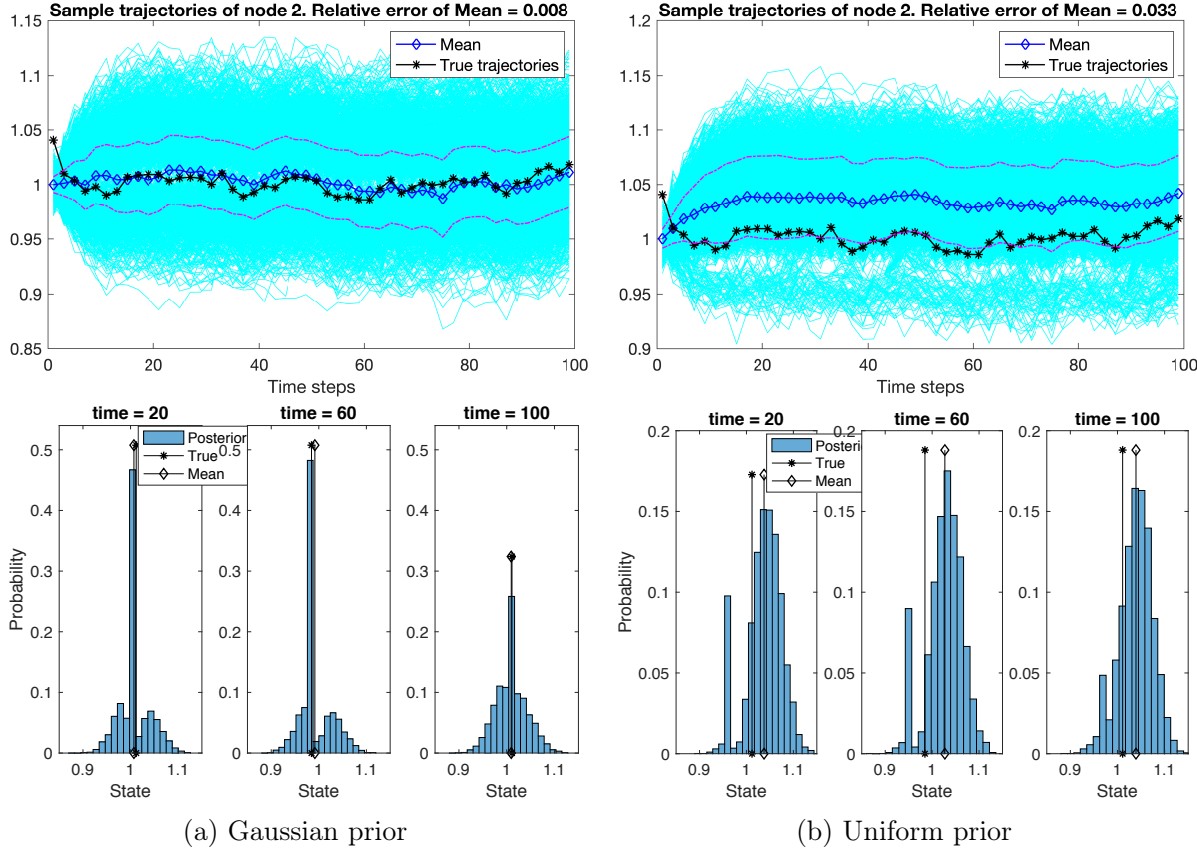

(a) Gaussian prior             (b) Uniform prior

**Figure 10.** The ensemble of sample trajectories of the state at an unobserved node. The ensembles exhibit a large uncertainty in both cases of priors, but the posterior means achieve relative errors of 0.8% and 3.3% in cases of Gaussian and uniform priors respectively. The one-standard-deviation band covers the true trajectory at most times. Bottom row: the histogram of samples at three time instants, showing that the samples concentrate around the true states. Particularly, in the case of the Gaussian prior, the peaks of the histogram are close to the true states, even when the histograms form a multi-mode distribution.

trajectories, superimposed on the ensembles of all sample trajectories at these nodes. We also present histograms of samples at three instants of time: $t = 20$, $t = 60$ and $t = 100$.

Figure 9 shows that the trajectory of the observed node is well estimated by the ensemble mean, with a relative error of 0.7%. Recall that the observation noise leads to a relative error of about 1%, so the posterior filters out 30% of the noise. Also note that the ensemble quantifies the uncertainty of the estimation, with the true trajectory being mostly enclosed within a one-standard-deviation band around the ensemble mean. Further, the histograms of samples at the three time instants show that the ensemble generally concentrates near the truth. In the Gaussian prior case, the peak of the histogram decreases as time increases. partially due to the degeneracy of SMC when we trace back the particles in time. In the uniform prior case, the ensembles are less concentrated than those in the Gaussian case, due to the wide spread of the parameter samples (Figure 7).



Figure 10 shows sample trajectories of an unobserved node. Despite the fact that the node is unobserved, the posterior means have relative errors of 0.8% and 3.3% in cases of Gaussian and uniform priors respectively, with a one-standard-deviation band covering the true trajectory at most times. While the sparse observations do cause large uncertainties for both priors, the histograms of samples show that the ensembles concentrate near the truth. Particularly, in the case of Gaussian prior, the peaks

of the histogram are close to the true states, even when the histograms form a multi-modal distribution due to the degeneracy of SMC.

We find that the posterior is able to filter out the noise in the observed nodes and reduce the uncertainty in the unobserved nodes from the climatological distribution. In particular, in the case of the Gaussian prior, the ensemble of posterior samples concentrates near the true state at both observed and unobserved nodes and substantially reduces the uncertainty. In the case of

the uniform prior, the ensemble of posterior samples spreads more widely, and only slightly reduces the uncertainty.

The coverage probability (CP), the proportion of the states whose 90% credible intervals contain the true values, is 95% in the Gaussian prior case and 92% for the uniform prior in the above simulation. The target probability is 90% as in this case 90% of the true values would be covered by 90% credible intervals. The values indicate statistically meaningful uncertainty estimates, for example larger uncertainty ranges at nodes with higher mean errors. The slight over-dispersiveness, i.e. higher

CPs than the target probabilities, might be a result of the large uncertainty in the parameter estimates.

Table 6 shows the means and standard deviations of the relative errors and CPs in state estimation by the posterior mean in 100 independent simulations, averaging over observed and unobserved notes. The relative errors at each time $t$ are computed by averaging the error of the ensemble mean (relative to the true value) over all the nodes. The relative error of the trajectory is the average over all times along the trajectory. The relative errors are 1.14% and 2.39% respectively for the cases of Gaussian

and uniform prior. These numbers are a result of averaging over the observed and unobserved nodes. Note that the relative errors are similar at different times $t = (20, 60, 100)$, indicating that the MCMC is able to ameliorate the degeneracy of the SMC to faithfully sample the posterior of the states.

In the Gaussian prior case, the CPs are above the target probability in the 100 independent simulations with a mean of 96%. This supports the finding from above that the posteriors are slightly over-dispersive due to the large uncertainty in the

parameter estimates. The standard deviation is very small with 2% which indicates the robustness of the Gaussian prior model. In the uniform prior case, the CPs are much lower with a mean of 73%. This might be a result of larger biases compared to the Gaussian prior case which are not compensated by larger uncertainty estimates. In addition, the standard deviation is much higher in the uniform prior case with 31%. This shows that this case is less robust than the Gaussian prior case.

## 5  Discussion

### 5.1  Observing fewer nodes

We tested the consequences of having sparser observations in space, e.g. observing only two out of the 12 nodes. In the Gaussian prior case, in a typical simulation with the same true parameters and observation data as in Section 4.2, the relative error in state estimation increases slightly, from 0.7% to 0.8% for the observed node and from 0.8% to 1.1% for the unobserved





**Table 6.** Means and standard deviations of the relative errors of the posterior mean trajectories of all nodes and the relative errors at three instants of time, computed from 100 independent simulations. In the last column, the mean and standard deviations of CPs are given in percent.

**(a) The case of observing six out of the 12 nodes.**

|  | Trajectory | $t = 20$ | $t = 60$ | $t = 100$ | CP |
|---|---|---|---|---|---|
| Gaussian Prior (%) | $1.14 \pm 0.41$ | $1.11 \pm 0.47$ | $1.09 \pm 0.47$ | $1.07 \pm 0.46$ | $96 \pm 2$ |
| Uniform Prior (%) | $2.39 \pm 1.59$ | $2.44 \pm 1.64$ | $2.42 \pm 1.66$ | $2.41 \pm 1.63$ | $73 \pm 31$ |

**(b) The case of observing two out of the 12 nodes.**

|  | Trajectory | $t = 20$ | $t = 60$ | $t = 100$ | CP |
|---|---|---|---|---|---|
| Gaussian Prior (%) | $1.43 \pm 0.44$ | $1.38 \pm 0.53$ | $1.43 \pm 0.51$ | $1.33 \pm 0.54$ | $92 \pm 6$ |
| Uniform Prior (%) | $2.46 \pm 1.28$ | $2.47 \pm 1.35$ | $2.49 \pm 1.33$ | $2.47 \pm 1.34$ | $75 \pm 25$ |

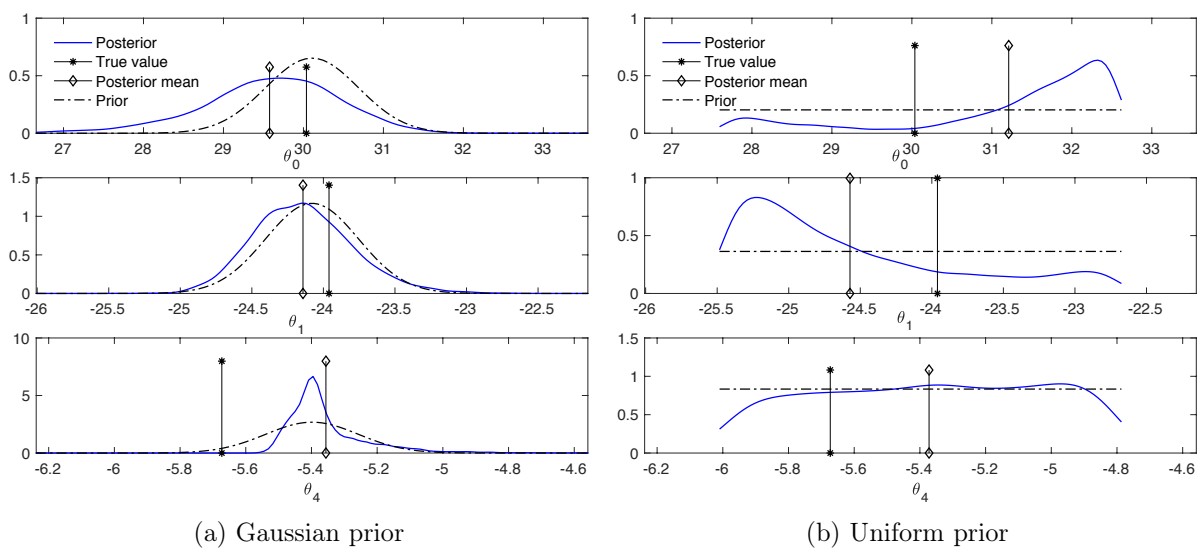

(a) Gaussian prior          (b) Uniform prior

**Figure 11.** The case of observing 2 out of the 12 nodes: marginal posteriors of $\theta$. With the same true parameters and the same observation dataset as in Figure 7, the marginal posteriors have slightly wider supports.

node. As a result, the overall error increases. The parameter estimates show small but noticeable changes (see Figure 11): the posteriors of the parameters have slightly wider support and the posterior means and MAPs exhibit slightly larger errors than those in Section 4.2.

We also ran 100 independent simulations to investigate sampling variability in the state and parameter estimates. Table 6(b) reports the means and standard deviations of the relative errors of the posterior mean trajectory, and CPs for state estimation in these simulations. The Gaussian prior case shows small increases in both the means and the standard deviations of errors, as well as slightly lower and less robust CPs. This confirms the results quoted above for a typical simulation. The uniform prior case shows almost negligible error and CP increases. Table 5(b) reports the mean and standard deviations of the posterior



means and MAP for parameter estimation in these simulations. Small changes in comparison to the results in Table 5(a) are found. These small changes are due to the strong regularization that has been introduced to overcome the degeneracy of the likelihood.

## 5.2 Observing a longer trajectory.

When the length $N$ of the trajectory of observation increases, the exponent of the regularized posterior (19), viewed as a function of $\theta$ only, tends to its expectation with respect to the ergodic measure of the system, i.e. $\frac{1}{N} C_{y_{1:N}}(\theta, u_{1:N}) \xrightarrow{N \to \infty} \mathbb{E}[C_{y_{1:N}}(\theta, u_{1:N})]$ almost surely. As a result, the marginal posterior tends to be stable as $N$ increases. This result indicates that an increase of data size has a limited effect on the regularized posterior of parameters. This fact is verified by numerical tests with $N = 1000$, in which the marginal posteriors only have a slightly wider support than those in Figure 7 with $N = 100$.

In general, the number of observations needed for the posterior to reach a steady state depends on the dimension of the parameters and the speed of convergence to the ergodic measure of the system. Here we have only three parameters and the SEBM converges to its stationary measure exponentially (in fewer than 10 time steps), therefore $N = 100$ is large enough to make the posterior be close to the steady state.

When the trajectory is long, a major issue is the computational cost from sampling the posterior of the states. Note that as $N$ increases, the dimension of the states in the posterior increases, demanding a longer Markov chain to explore the target distribution. In numerical tests with $N = 1000$, the correlation length of the Markov chain is at least 100, about four times the correlation length found for $N = 100$. Therefore, to obtain the same number of effective samples as before, we would need a Markov chain with length at least four times the previous length, say, $L = 4 \times 10^4$. The computational cost increases linearly in $NL$, with each step requiring an integration of the SPDE. The high computational cost, an instance of the well-known "curse of dimensionality", renders the direct sampling of the posterior unfeasible. Two groups of methods could reduce the computational cost and make the Bayesian inference feasible. The first group of methods, dynamical model reduction, exploits the low-dimensional structure of the stochastic process to develop low-dimensional dynamical models which efficiently reproduce the statistical-dynamical properties needed in the SMC (see e.g. Chorin and Lu, 2015; Lu et al., 2017; Chekroun and Kondrashov, 2017; Khouider et al., 2003, and the references therein). The other group of methods approximates the marginal posterior of the parameter by reduced order models for the response of the data to parameters (see e.g. Marzouk and Najm, 2009; Branicki and Majda, 2013; Cui et al., 2015; Chorin et al., 2016; Lu et al., 2015; Jiang and Harlim, 2018)). In a paleoclimate reconstruction context, the number of observations will generally be determined by available observations and the length of the reconstruction period rather than by computational considerations. We leave these further development of efficient sampling methods for long trajectories as a direction of future research.

## 5.3 Estimates of the nonlinear function

One goal of parameter estimation is to identify the nonlinear function $g_\theta$ (specified in (2)) in the SEBM. The posterior of the parameters also quantifies the uncertainty in the identification of $g_\theta$. Figure 12 shows the nonlinear function $g_\theta$ associated with the true parameters and with the MAPs and posterior means presented in Figure 7, superposed on an ensemble of the nonlinear





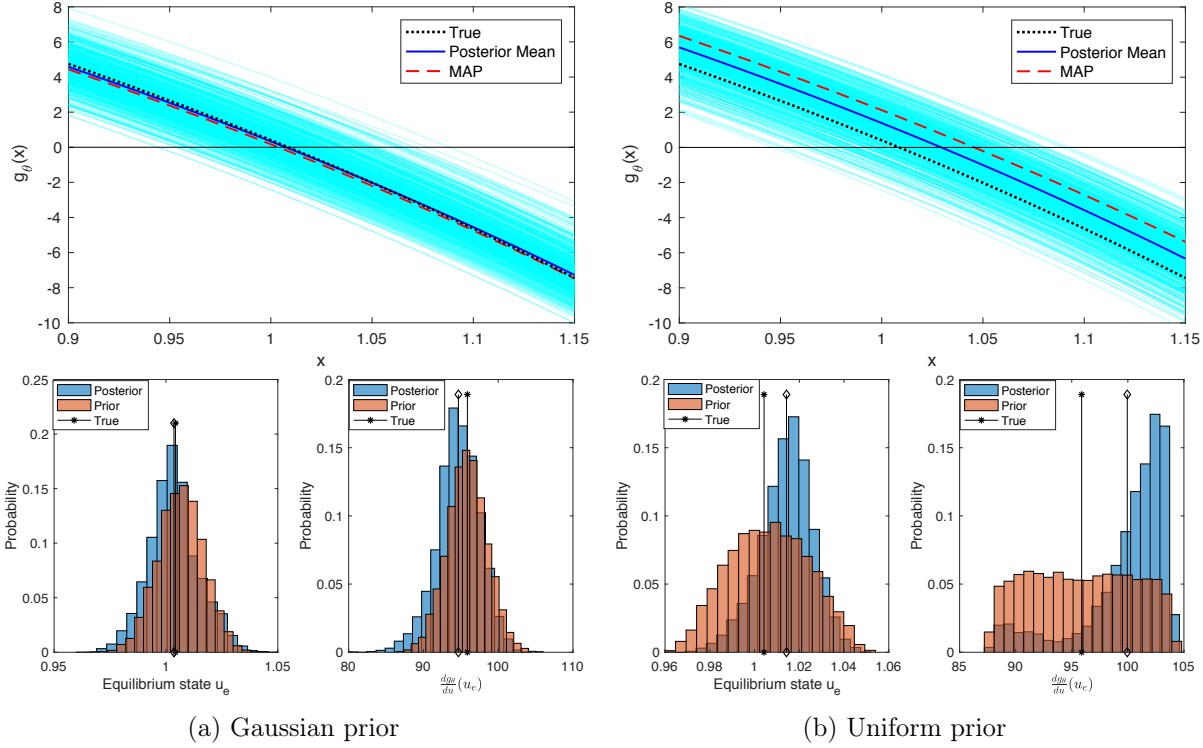

(a) Gaussian prior          (b) Uniform prior

**Figure 12.** *Top row:* The true nonlinear function $g_\theta$ and its estimators using posterior mean and MAP, superposed on the ensemble of all estimators using the samples. *Bottom row:* The distribution of the equilibrium state $u_e$ (i.e. the zero of the nonlinear function $g_\theta(\cdot)$) and the distribution of $\frac{dg_\theta}{du}(u_e)$, with $\theta$ being samples of the prior and of the posterior.

function evaluated with all the samples. Note that in the Gaussian prior case, the true and estimated functions $g_\theta$ are close even though $\theta_4$ is estimated with large biases by either the posterior mean or by the MAP. In the uniform prior case, the posterior mean has a smaller error than the MAP and leads to a better estimate of the nonlinear function. In either case, the large band of the ensemble represents a large uncertainty in the estimates.

5      For the Gaussian prior, neither the posterior distribution of the equilibrium state $u_e$ (for which $g_\theta(u_e) = 0$) nor of the feedback strength $dg_\theta/du(u_e)$ are substantially changed from the corresponding priors. Both experience only a small reduction of uncertainty. In contrast, the posterior distributions are narrower than the priors for the uniform prior case - although the posterior means and MAPs are both biased.

## 6    Conclusions and future work

10      We have investigated the joint state-parameter estimation of a nonlinear stochastic energy balance model (SEBM) motivated by the problem of spatial-temporal paleoclimate reconstruction from sparse and noisy data, for which parameter estimation is an ill-posed inverse problem. We introduced strongly regularized posteriors to overcome the ill-posedness by restricting the





parameters and states to physical ranges and by normalizing the likelihood function. We considered both a uniform prior and a more informative Gaussian prior based on the physical ranges of the parameters. We sampled the regularized high-dimensional posteriors by a Particle Gibbs with Ancestor Sampling (PGAS) sampler that combines Markov Chain Monte Carlo (MCMC) with an optimal particle filter to exploit the forward structure of the SEBM.

Results show that the regularization overcomes the ill-posedness in parameter estimation and leads to physical posteriors quantifying the uncertainty in parameter-state estimation. Due to the ill-posedness, the posterior of the parameters features a relatively large uncertainty. This result implies that there can be a large uncertainty in point estimators such as the posterior mean or the maximum a posteriori (MAP), the latter of which corresponds to the minimizer in a variational approach with regularization. Despite the large uncertainty in parameter estimation, the marginal posteriors of the states generally concentrate

near the truth, reducing the uncertainty in state reconstruction. In particular, the more informative Gaussian prior leads to much better estimations than the uniform prior: the uncertainty in the posterior is smaller, the MAP and posterior mean have smaller errors in both state and parameter estimates, and the coverage probabilities are higher and more robust.

Results also show that the regularized posterior is robust to spatial sparsity of observations, with sparser observations leading to slightly larger uncertainties due to less information. However, due to the need of regularization to overcome ill-posedness,

the uncertainty in the posterior of the parameters cannot be eliminated by increasing the number of observations in time. Therefore, we suggest alternative approaches, such as re-parametrization of the nonlinear function according to the climatological distribution or nonparametric Bayesian inference (see e.g. Müller and Mitra, 2013; Ghosal and Van der Vaart, 2017) to avoid ill-posedness.

The ill-posedness of the parameter estimation problem for the model we have considered is of particular interest because the

form of the nonlinear function $g_\theta(u)$ is not arbitrary but is motivated by the physics of the energy budget of the atmosphere. The fact that wide ranges of the parameters $\theta_i$ are consistent with the "obserations" even in this highly idealized setting indicates that surface temperature observations themselves may not be sufficient to constrain physically-important parameters such as albedo, graybody thermal emissivity, or air-sea exchange coefficients separately. While state-space modeling approaches allow reconstruction of past surface climate states, it may be the case that the associated climate forcing may not contain sufficient

information to extract the relative contributions of the individual physical processes that produced it.

## Appendix A: Technical details of the estimation procedure

### A1   Discretization of the SEBM

**Finite element representation in space** We discretize the SEBM in space by finite element methods (see e.g. Alberty et al., 1999). Denote by $\{\phi_i(\xi)\}_{i=1}^{d_b}$ the finite element basis functions, and approximate the solution $u(t,\xi)$ by

$$u_{d_b}(t,\xi) = \sum_{i=1}^{d_b} \widehat{u}_i(t)\phi_i(\xi). \tag{A1}$$



The coefficients $\widehat{u}_i$ are determined by the following weak Galerkin projection of the SEBM (1)

$$\langle u_{d_b}(t,\cdot),\phi\rangle = \langle u_0,\phi\rangle - \nu\int_0^t \langle\nabla u_{d_b}(s,\cdot),\nabla\phi\rangle ds + \int_0^t\langle g_\theta(u_{d_b}(s,\cdot)),\phi\rangle ds + \int_0^t\langle f(s,\cdot),\phi\rangle, \tag{A2}$$

where $\phi$ is a continuously differentiable compactly supported test function and the integral $\int_0^t\langle f(s,\cdot),\phi\rangle$ is an Itô integral.

For convenience, we write this Galerkin approximate system in vector notation. Denote

$$U(t) = (\widehat{u}_1(t),\ldots,\widehat{u}_{d_b}(t))^T, \tag{A3}$$

$$\Phi(\xi) = (\phi_1(\xi),\ldots,\phi_{d_b}(\xi))^T, \tag{A4}$$

$$u_{d_b}(t,\xi) = U^T(t)\Phi(x) = \Phi^T(x)U(t). \tag{A5}$$

Taking $\phi = \phi_j$, $j = 1,\ldots,d_b$ in equation (A2) and using the symmetry of the inner product, we obtain a stochastic integral equation for the coefficient $U(t) \in \mathbb{R}^{d_b}$:

$$\langle\Phi,\Phi^T\rangle U(t) = \langle\Phi,\Phi^T\rangle U(0) - \nu\langle\nabla\Phi,\nabla\Phi^T\rangle\int_0^t U(s)\,ds + \int_0^t\langle g_\theta(U_n^T\Phi),\Phi\rangle\,ds + \int_0^t\langle f(s,\cdot),\Phi\rangle. \tag{A6}$$

To simplify notation, we denote the mass and stiffness matrices by

$$\mathbf{M}_0 = \langle\Phi,\Phi^T\rangle, \qquad \mathbf{M}_1 = \nu\langle\nabla\Phi,\nabla\Phi^T\rangle, \tag{A7}$$

which are symmetric, tri-diagonal, positive definite matrices in $\mathbb{R}^{d_b\times d_b}$, and we denote the nonlinear term as

$$G_\theta(U(t)) := \langle g_\theta(U^T(t)\Phi),\Phi\rangle. \tag{A8}$$

The above stochastic integral equation can then be written as

$$\mathbf{M}_0 U(t) = \mathbf{M}_0 U(0) - \mathbf{M}_1\int_0^t U(s)\,ds + \int_0^t G_\theta(U(t))\,ds + \int_0^t\langle f(s,\cdot),\Phi\rangle. \tag{A9}$$

The mesh on the sphere and the matrices $\mathbf{M}_0$ and $\mathbf{M}_1$ are computed with the R package INLA (Lindgren and Rue, 2015; Bakka et al., 2018).

**Representation of the nonlinear term.** The parametric nonlinear functional $G_\theta(U(t))$ is approximated using the finite elements. We approximate each spatial integration over an element-triangle in $\langle g_\theta(U_n^T\Phi),\Phi\rangle$ by the volume of the triangular pyramid whose height is the value of the nonlinear function at the center of the element-triangle $\mathcal{T}_k$, i.e.

$$\int g_\theta(u(t,\xi))\phi_l(\xi)d\xi \approx \sum_{\mathcal{T}_k\subset\mathrm{supp}(\phi_l)}\frac{\mathrm{Area}(\mathcal{T}_k)}{3}g_\theta\left(\sum_i U_i(t)\phi_i(\xi_k^c)\right), \tag{A10}$$

where $\xi_k^c$ is the center of the triangle $\mathcal{T}_k$. In the discretized system, we assume that this approximation has a negligible error and take it as our nonlinear functional. In vector notation, it reads

$$G_\theta(U(t)) = A_\mathcal{T}g_\theta(\mathbf{A}U(t)), \tag{A11}$$





where $A_\mathcal{T} = \left( \frac{\text{Area}(\mathcal{T}_k)}{3} \right) \in \mathbb{R}^{d_b \times d_e}$ with $d_e$ denoting the number of triangle elements and the matrix $\mathbf{A} = (\phi_i(\xi_k^c)) \in \mathbb{R}^{d_e \times d_b}$, such that the function $g_\theta(\mathbf{A}U(t))$ is interpreted as element-wise evaluation. For the nonlinear function $g_\theta$ in (2), we can write the above nonlinear term as

$$G_\theta(U(t)) = \sum_{k=0,1,4} \theta_k A_\mathcal{T}(\mathbf{A}U(t))^{\circ k}, \tag{A12}$$

where $\circ k$ denotes entry-wise product of the array.

**Representation of the stochastic forcing.** Following Lindgren et al. (2011), the stochastic forcing $f(t,\xi)$ is approximated by its linear finite element truncation,

$$f(t,\xi) = \sum_{i=1}^{d_b} \phi_i(\xi) f_i(t) \tag{A13}$$

with the stochastic processes $\{f_i(t), i=1,\ldots,d_b\}$ being spatially correlated and white in time. Note that for $\nu = 0.1$ and $\rho > 0$ in the Matérn covariance (4), the process $f(t,\xi)$ is the stationary solution of the stochastic Laplace equation

$$(\rho^{-2} - \nu \triangle)f(t,\xi) = \sigma_f W(t,\xi), \tag{A14}$$

where $W$ is a spatio-temporal white noise (Whittle, 1954, 1963). Computationally efficient approximations of the forcing process are obtained using the GMRF approximation of Lindgren et al. (2011) which generates $F(t) \equiv (f_1(t), f_2(t), \ldots, f_{d_b}(t))$ by solving (A14). That is, using the above finite element notation, we solve for each time $t$ the linear system

$$\left( \rho^{-2}\mathbf{M}_0 + \mathbf{M}_1 \right) F(t) = \sigma_f \langle \Phi, W(t,\cdot) \rangle, \tag{A15}$$

where the random vector $\langle \Phi, W(t,\cdot) \rangle := (\langle \phi_1, W(t,\cdot) \rangle, \ldots, \langle \phi_{d_b}, W(t,\cdot) \rangle)$ is Gaussian with mean 0 and covariance $\mathbf{M}_0$. Solving (A15) yields

$$F(t) \sim \mathcal{N}\left( 0, \sigma_f^2 \mathbf{M}_\rho^{-1} \mathbf{M}_0 \mathbf{M}_\rho^{-1} \right), \tag{A16}$$

where $\mathbf{M}_\rho := (\rho^{-2}\mathbf{M}_0 + \mathbf{M}_1)$.

**Semi-backward Euler time integration.** Equation (A9) is integrated in time by a semi-backward Euler scheme

$$\mathbf{M}_{\Delta t} U_{n+1} = \mathbf{M}_0 U_n + \Delta t\, G_\theta(U_n) + \sqrt{\Delta t}\, \mathbf{M}_0 F_n, \tag{A17}$$

where $U_n$ is the approximation of $U(t_n)$ with $t_n = n\Delta t$, and $\{F_n\}$ is a sequence of iid random vectors with distribution $\mathcal{N}\left( 0, \sigma_f^2 \mathbf{M}_\rho^{-1} \mathbf{M}_0 \mathbf{M}_\rho^{-1} \right)$, with the matrix $\mathbf{M}_{\Delta t}$ denoting

$$\mathbf{M}_{\Delta t} := \mathbf{M}_0 + \Delta t\, \mathbf{M}_1. \tag{A18}$$

**Efficient generation of the Gaussian field.** It follows from (A15) that $\mathbf{M}_0 F_n$ is Gaussian with mean zero and covariance $\mathbf{M}_0 \mathbf{M}_\rho^{-1} \mathbf{M}_0 \mathbf{M}_\rho^{-1} \mathbf{M}_0$. Note that while $\mathbf{M}_\rho$ is a sparse matrix, its inverse matrix $\mathbf{M}_\rho^{-1}$ is not. To efficiently use the sparseness of $\mathbf{M}_\rho$, following Lindgren et al. (2011), we approximate $\mathbf{M}_0$ by $\widehat{\mathbf{M}}_0 := \text{diag}(\langle \phi_i, 1 \rangle)$ and compute the noise $\mathbf{M}_0 F_n$





by $\mathbf{C}^{-1}\mathcal{N}(0, I_d)$, where $\mathbf{C}$ is the Cholesky factorization of the inverse of the covariance matrix (called precision matrix) $\widehat{\mathbf{M}}_0^{-1}\mathbf{M}_\kappa\widehat{\mathbf{M}}_0^{-1}\mathbf{M}_\kappa\widehat{\mathbf{M}}_0^{-1}$. The precision matrix is a sparse representation of the inverse of the covariance. Therefore, the matrix $\mathbf{C}$ is also sparse and the noise sequence can be efficiently generated.

In summary, we can write the discretized SEBM in the form

$$U_{n+1} = \mu_\theta(U_n) + W_n \tag{A19}$$

where the deterministic function $\mu_\theta(\cdot)$ is given by

$$\mu_\theta(U_n) = \mathbf{M}_{\Delta t}^{-1}\mathbf{M}_0 U_n + \sum_{k=0,1,4} \theta_k G_{\theta,k}(U_n), \tag{A20}$$

with $G_{\theta,k}(U_n) := \Delta t \mathbf{M}_{\Delta t}^{-1} A_{\mathcal{T}}(\mathbf{A}U(t))^{\circ k}$, and $\{W_n\}$ is a sequence of iid Gaussian noise with mean 0 and covariance $\mathbf{R}$:

$$\mathbf{R} = \sigma_f^2 \Delta t \mathbf{M}_{\Delta t}^{-1}\mathbf{C}^{-1}\mathbf{C}^{-T}\mathbf{M}_{\Delta t}^{-T}. \tag{A21}$$

## A2 SMC with optimal importance sampling

SMC methods approximate the target density $p_\theta(u_{1:N}|y_{1:N})$ sequentially by weighted random samples called particles (hereafter we drop the subindex $\theta$ to simplify notation)

$$\widehat{p}(u_{1:N}|y_{1:N}) := \sum_{m=1}^{M} w_n^m \delta_{U_{1:n}^m}(du_{1:N}). \tag{A22}$$

with $\sum_{m=1}^{M} w_n^m = 1$. These weighted samples are drawn sequentially by importance sampling based on the recurrent formation

$$p(u_{1:n}|y_{1:n}) = p(u_{1:n-1}|y_{1:n-1})\frac{p(y_n|u_n)p(u_n|u_{n-1})}{p(y_n|y_{1:n-1})}. \tag{A23}$$

More precisely, suppose that at time $n$, we have weighted samples $\{U_{1:n-1}^m, w_{n-1}^m\}_{m=1}^M$. One first draws a sample $U_n^m$ from an easy to sample importance density $q(u_n|y_n, U_{n-1}^m)$ that approximates the "incremental density" which is proportional to $p(y_n|u_n)p(u_n|U_{n-1}^m)$ for each $m = 1, \ldots, M$, and computes incremental weights

$$\alpha_n^m = \frac{p(U_n^m|U_{n-1}^m)p(y_n|U_n^m)}{q(U_n^m|y_n, U_{n-1}^m)}, \tag{A24}$$

which account for the discrepancy between the two densities. One then assigns normalized weights $\{w_n^m \propto w_{n-1}^m \alpha_n^m\}_{m=1}^M$ to the concatenated sample trajectories $\{U_{1:n}^m\}_{m=1}^M$.

A clear drawback of the above procedure is that all but one of the weights $\{w_n^m\}$ will become close to zero as the number of iterations increases, due to the multiplication and normalization operations. To avoid this, one replaces the unevenly weighted samples $\{(U_{n-1}^m, w_{n-1}^m)\}$ by uniformly weighted samples from the approximate density $\widehat{p}_\theta(u_{n-1}|y_{1:N-1})$. This is the well-known *resampling* technique. In summary, the above operations are carried out as follows:





(i) draw random indices $\{A_{n-1}^m\}_{m=1}^M$ according to the discrete probability distribution $\mathbb{F}(\cdot|w_{n-1}^{1:M})$ on the set $\{1,\ldots,M\}$, which is defined as

$$\mathbb{F}(A_{n-1} = k|w_{n-1}^{1:M}) = w_{n-1}^k, \text{ for } k = 1,\ldots,M. \tag{A25}$$

(ii) for each $m$, draw a sample $U_n^m$ from $q(u_n|y_n, U_{n-1}^{A_{n-1}^m})$ and set $U_{1:n}^m := (U_{n-1}^{A_{n-1}^m}, U_n^m)$;

(iii) compute and normalize the weights

$$\alpha_n^m := \alpha_n(U_{1:n}^m) = \frac{p(U_n^m|U_{n-1}^{A_{n-1}^m})p(y_n|U_n^m))}{q(U_n^m|y_n, U_{n-1}^{A_{n-1}^m})}, \quad w_n^m := \frac{\alpha_n^m}{\sum_{k=1}^M \alpha_n^k}. \tag{A26}$$

The above SMC sampling procedure is called sequential importance sampling with resampling (SIR) (Doucet and Johansen, 2011, see e.g.[]) and is summarized in Algorithm 1.

---

**Algorithm 1** Sequential importance sampling with resampling (SIR).

---

**Require:** Observation $y_{1:N}$ and ensemble size $M$. For the SEBM, we use the optimal importance density $q$ in (A27). Each step is for $m = 1,\ldots,M$.

**Ensure:** Weighted samples $\{(U_{1:N}^m, w_N^m)\}_{m=1}^M$.

1: Draw samples $U_1^m \sim q(u_1|y_1)$.

2: Compute and normalize the weights: $\alpha_1^m = \frac{p_\theta(U_1^m)p_\theta(y_1|U_1^m))}{q(U_1^m|y_1)}$, $w_1^m = \frac{\alpha_1^m}{\sum_{k=1}^M \alpha_1^k}$.

3: **for** $n = 2 : N$ **do**

4:    Draw samples $A_{n-1}^m \sim \mathbb{F}(\cdot|w_{n-1}^{1:M})$ with $\mathbb{F}$ defined in (A25).

5:    Draw samples $U_n^m \sim q(u_n|y_n, U_{n-1}^{A_{n-1}^m})$ and set $U_{1:n}^m := (U_{n-1}^{A_{n-1}^m}, U_n^m)$.

6:    Compute the normalized weights $w_n^m$ according to (A26).

7: **end for**

---

**Optimal importance sampling.** Note that the conditional transition density of the states $p_\theta(u_{n+1}|u_n)$ in (7) is Gaus-
sian and the observation model in (8) is linear and Gaussian. These facts allow for a Gaussian optimal importance density $q(u_n|y_n, U_{n-1}^m)$ that is proportional to $p(y_n|u_n)p(u_n|U_{n-1}^m)$ for each $m = 1,\ldots,M$:

$$q(u_n|y_n, U_{n-1}^m) \sim \mathcal{N}(\mu_n^m, \mathbf{\Sigma}) \tag{A27}$$

with the mean $\mu_n^m$ and the covariance $\mathbf{\Sigma}$ given by

$$\mu_n^m = \mu(U_{n-1}^m) + \mathbf{R}\mathbf{H}^T\mathbf{Q}^{-1}(y_n - \mathbf{H}\mu(U_{n-1}^m)), \tag{A28}$$

$$\mathbf{\Sigma} = \mathbf{R} - \mathbf{R}\mathbf{H}^T\left(\mathbf{Q} + \mathbf{H}\mathbf{R}\mathbf{H}^T\right)^{-1}\mathbf{H}\mathbf{R}. \tag{A29}$$

**Drawbacks of SMC.** While the resampling technique prevents $w_n^m$ from being degenerate at each current time $n$, SMC algo-
rithms suffer from the degeneracy (or particle depletion) problem: the marginal distribution $\widehat{p}(u_n|(y_{1:N}))$ becomes concentrated
on a single particle as $N - n$ increases because each resampling step reduces the number of distinct particles of $u_n$. As a result,
the estimate of the joint density $p(u_{1:N}|y_{1:N})$ of the trajectory deteriorates as time $N$ increases.





## A3 Particle Gibbs and PGAS

The framework of particle MCMC introduced in Andrieu et al. (2010) is a systematic combination of SMC and MCMC methods, exploiting the strengths of both techniques. Among the various particle MCMC methods, we focus on the *particle Gibbs sampler* (PG) that uses a novel conditional SMC update (Andrieu et al., 2010), as well as its variant, the *particle Gibbs*

*with ancestor sampling* (PGAS) sampler (Lindsten et al., 2014), because they are best fit for sampling our joint parameter and state posterior.

The PG and PGAS samplers use a conditional SMC update step to realize the transition between two steps of the Markov chain while ensuring that the target distribution will be the stationary distribution of the Markov chain. The basic procedure of a PG sampler is as follows:

– Initialization: draw $\theta(1)$ from the prior distribution $p(\theta)$. Run an SMC algorithm to generate weighted samples $\{U_{1:N}^m, w_N^m\}_{m=1}^M$ for $p_{\theta(1)}(u_{1:N}|y_{1:N})$ and draw $U_{1:N}(1)$ from these weighted samples.

– Markov chain iteration: for $l = 1, \cdots, L-1$,

1. Sample $\theta(l+1)$ from the marginal posterior $p(\theta|y_{1:N}, U_{1:N}(l))$ given by (14).

2. Run a *conditional SMC algorithm*, conditioned on $U_{1:N}(l)$, which is called the reference trajectory. That is, in the

SMC algorithm, the $M$-th particle is required to move along the reference trajectory by setting $U_n^M = U_n(l)$. Draw other samples from the importance density, and normalize the weights and resample all the particles as usual. This leads to weighted samples $\{U_{1:N}^m, w_N^m\}_{m=1}^M$ with $U_{1:N}^M = U_{1:N}(l)$.

3. Draw $U_{1:N}(l+1)$ from the above weighted samples.

– Return the Markov chain $\{\theta(l), U_{1:N}(l)\}_{l=1}^L$.

The conditional SMC algorithm is the core of PG samplers. It retains the reference path throughout the resampling steps by deterministically setting $U_{1:N}^M = U_{1:N}(l)$ and $A_n^M = M$ for all $n$, while sampling the remaining $M-1$ particles according to a standard SMC algorithm. The reference path interacts with the other paths by contributing a weight $w_n^M$. This is the key to ensuring that the PG Markov chain converges to the target distribution. A potential risk of the PG sampler is that it yields a poorly mixed Markov chain, because the reference trajectory tends to dominate the SMC ensemble trajectories.

The PGAS sampler increases the mixing of the chain by connecting the reference path to the history of other particles by assigning an ancestor to the reference particle at each time. This is accomplished by drawing a sample for the ancestor index $A_{n-1}^M$ of the reference particle, which is referred to as *ancestor sampling*. The distribution of the index $A_{n-1}^M$ is determined by the likelihood of connecting $U_n(l)$ to the particles $\{U_{n-1}^m\}_{m=1}^M$, in other words, according to weights

$$\widetilde{\alpha}_{n-1|n}^m = w_{n-1}^m p_{\theta(l+1)}(U_n(l)|U_{n-1}^m) p(y_n|U_n(l)),$$

$$\widetilde{w}_{n-1|n}^m = \frac{\widetilde{\alpha}_{n-1}^m}{\sum_{k=1}^M \widetilde{\alpha}_{n-1}^k} \tag{A30}$$





The above weight $\widetilde{\alpha}_{n-1|n}^m$ can be seen as a posterior probability, where the importance weight $w_{n-1}^m$ is the prior probability of the particle $U_{n-1}^m$, and the product $p_{\theta(l+1)}(U_n(l)|U_{n-1}^m)p(y_n|U_n(l))$ is the likelihood that $U_n(l)$ originates from $U_{n-1}^m$ conditional on observation $y_n$. In short, the PGAS sampler assigns the reference particle $U_n(l)$ an ancestor $A_{n-1}^M$ that is drawn from the distribution $\mathbb{F}(A_{n-1}^M = k|\widetilde{w}_{n-1|n}^{1:M}) = \widetilde{w}_{n-1|n}^k$.

5    The above conditional SMC with ancestor sampling within PGAS is summarized in Algorithm 2.

---

**Algorithm 2** Conditional SMC with ancestor sampling for PGAS sampler.

---

**Require:** $U_{1:N}(l)$ and $\theta := \theta(l+1)$.

**Ensure:** $U_{(1:N)}(l+1)$.

   Initialize the particles in SMC:

1:    Set $U_1^M = U_1(l)$ and draw samples $\{U_1^m\}_{m=1}^{M-1} \sim q_\theta(x_1|y_1)$.

2:    Compute the weights $\alpha_1^m = \frac{p_\theta(U_1^m)p_\theta(y_1|U_1^m))}{q_\theta(U_1^m|y_1)}, w_1^m = \frac{\alpha_1^m}{\sum_{k=1}^M \alpha_1^k}$ for $m = 1 : M$.

3: **for** $n = 2 : N$ **do**

4:    Draw samples $\{A_{n-1}^m\}_{m=1}^{M-1} \sim \mathbb{F}(\cdot|w_{n-1}^{1:M})$.

5:    Set $U_n^M = U_n(t)$ and draw samples $U_n^m \sim q(x_n|y_n, U_{n-1}^{A_{n-1}^m})$ for $m = 1 : M - 1$.

6:    Draw $A_{n-1}^M \sim \mathbb{F}(\cdot|\widetilde{w}_{n-1|n}^{1:M})$, where the weights in $\widetilde{w}_{n-1|n}^{1:M}$ are computed in (A30).

7:    Set $U_{1:n}^m := (U_{1:n-1}^{A_{n-1}^m}, U_n^m)$ for $m = 1 : M$.

8:    Compute the normalized weights $w_n^m$ according to (A26).

9: **end for**

10: Draw $A_N$ with $\mathbb{F}(\cdot|w_N^{1:M})$.

11: **return** $U_{(1:N)}(l+1) = U_{1:N}^{A_N}$.

---

*Author contributions.* FL, NW and AM formulated the project and designed the experiments. NW and AM derived the SEBM, FL and NW carried out the experiments. FL developed the model code and performed the simulations with contributions from NW. FL prepared the manuscript with contributions from all co-authors.

*Competing interests.* The authors declare that they have no conflict of interest.

10    *Disclaimer.* Any opinions, findings, and conclusions or recommendations expressed in this material are those of the author(s) and do not necessarily reflect the views of the National Science Foundation.

*Acknowledgements.* This research started in a working group supported by the Statistical and Applied Mathematical Sciences Institute (SAMSI). FL thanks Prof. Peter Jan van Leeuwen, Prof. Kayo Ide, Prof. Mauro Maggioni, Prof. Xuemin Tu, and Dr. Wenjun Ma for helpful discussions. FL is supported by the National Science Foundation under Grant DMS-1821211. NW thanks Andreas Hense and Douglas Ny-
15    chka for inspiring discussions. NW was supported by the German Federal Ministry of Education and Research (BMBF) through the Palmod project (FKZ: 01LP1509D). NW thanks the German Research Foundation (code RE3994-2/1) for funding. AM acknowledges support from the Natural Sciences and Engineering Research Council of Canada (NSERC), and thanks SAMSI for hosting him in the autumn of 2017.



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
