# Peer review of "Joint state-parameter estimation of a nonlinear stochastic energy balance model from sparse noisy data"

_Nonlinear Processes in Geophysics, 2019_

## Referee Comment (RC1) · Colin Grudzien (Referee) · 1 Jun 2019

GENERAL RESPONSE:

I want firstly thank the editor for selecting me to review this work, and for providing a short extension to the review period – my apologies for being slightly late with completing this review.

I found that this is of very high quality analysis and exposition, and it was quite a pleasure to read. In this regard, I don't have many suggestions for improvement of the text, only a few minor points that I think can be elaborated on. I think the work should

therefore be accepted for publication after the authors address the following points.

MINOR POINTS FOR REVISION:

1) Page 7, line 20: among the references mentioned for local particle filters, I think it is worth mentioning the recent review article (https://www.nonlin-processes-geophys.net/25/765/2018/), which has an up-to-date survey of different localization techniques, and a classification and comparison of methods therein.

2) Page 14, figure 5 caption: I believe there is a typo, where "notes" should be "nodes".

3) Page 20, lines 19 - 25: I think this discussion is very interesting and useful to the reader. I would like this to be expanded to elaborate on the consequences for the analysis of paleo-climate in terms of the contributions of different processes to past states, and how this may affect the inferences we may wish to make based on such models. Likewise, I would like this to elaborate further on the more realistic case of nonlinear observation operators relating proxy measurements to the actual climate system, and how the reconstruction of the climate and parameters will be affected by these additional complications. I think the work will benefit from a longer conclusion and discussion of the implications of the results for more realistic modeling settings.

4) Page 31, algorithm 2, line 5: I believe there is a typo in the statement $U_n^M = U_n(t)$ , should the "t" be replaced by an "l"?
* * *

---

## Referee Comment (RC2) · Anonymous Referee #2 · 13 Jun 2019

**General comments**

This paper discusses how to estimate parameters for spatial-temporal models. It considers testing the methodologies on a nonlinear stochastic energy balance model (SEBM) in paleoclimate reconstruction problems. To resolve the ill-posed Fisher information, it considers using a strongly regularized approach. Systematic numerical tests were run using the particle Gibbs ancestral sampler (PGAS).

Overall, I think this paper is well written, and it provides detailed discussions on the related methodologies. There are a few minor issues listed below. I recommend its
publication after these issues are addressed.

**Specific comments**

Paleoclimate reconstruction seems to be an interesting and nonstandard question. So it might be worthy to give some details on how the SEBM is formulated, for example, what do each parameter $\theta$ stands for. It is also better to give some references on why the specific parameters are chosen. For example, it seems that data are available with time-interval $\Delta = 0.01$ (what is the time unit here?). But is this practically true?

While I agree the Fisher information matrix may be ill-conditioned, but I don't see immediately why the strong regularization approach is the right or natural way to fix it. The numerical results show that the regularized posterior has obvious biases, and sometimes close to being the prior. The strong regularization used here might be the cause of this. An alternative approach might be using the following version instead of (21)

$$p^N(\theta|u, y) \propto p(\theta)^\alpha [p_\theta(u)]^{1/N}$$

where $\alpha$ is a parameter in $[0, 1]$, and it can be tuned for a better posterior.

**Technical corrections**

1. Page 9, Fisher information matrix: since most NPG readers are likely to be geoscientists, maybe you should explain that in statistics, Fisher information dictates the asymptotic inference difficulty and give references. Also, there should be some explanations on why this matrix is ill-conditioned, not just some simulation plots.
2. Figure 2. The figure caption below "Data size $\log_{10} N$" is garbled. This happens to many figures later on as well. This might be a problem with my own computer/printer. But you better check.

3. Page 11, what is $u_i$? And it is better to define $u_c$ and $\sigma_o$ with mathematical terms.

4. Page 13, I think Markov chain might be too abstract a term for NPG. I think you can replace it with MCMC for the same meaning.

5. Page 18, line 2, there shouldn't be parenthesis for $\theta_0, \theta_1$.

6. Page 29, line 7, "see e.g. [". You miss some content here.
* * *

---

## Author Comment (AC1) · 28 Jun 2019

Dear Dr. Grudzien,

We thank you for carefully reading the manuscript and for the valuable feedback.

We have added the references in your comment 1) and corrected the typos mentioned in your comments 2) and 4).

About your comment 3): "Page 21, lines 19 - 25: I think this discussion is very interesting and useful to the reader. I would like this to be expanded to elaborate on the consequences for the analysis of paleo-climate in terms of the contributions of different

processes to past states, and how this may affect the inferences we may wish to make based on such models. Likewise, I would like this to elaborate further on the more realistic case of nonlinear observation operators relating proxy measurements to the actual climate system, and how the reconstruction of the climate and parameters will be affected by these additional complications. I think the work will benefit from a longer conclusion and discussion of the implications of the results for more realistic modeling settings."

We extended our discussion of the implications of our results for paleoclimate reconstructions by adding a subsection '5.4 Implications for paleoclimate reconstructions'. This section covers steps to be taken towards more realistic observation operators and discusses the implications for reconstructing physically meaningful state estimates and inference of parameters for energy sinks and sources. In turn, we shortened the respective paragraph in the conclusions.

**5.4 Implications for paleoclimate reconstructions**
Our analysis shows that assessing the well-posedness of the inverse problem of parameter estimation is a necessary first step for paleoclimate reconstructions making use of physically motivated parametric models. When the problem is ill-posed, a straightforward Bayesian inference will lead to biased and unphysical parameter estimates. We overcome this issue by using regularized posteriors, resulting in parameter estimates in the physically reasonable range with quantified uncertainty. However, it should be kept in mind that this approach relies strongly on high quality prior distributions.

The ill-posedness of the parameter estimation problem for the model we have considered is of particular interest because the form of the nonlinear function $g_\theta(u)$ is not arbitrary but is motivated by the physics of the energy budget of the atmosphere. The fact that wide ranges of the parameters $\theta_i$ are consistent with the "observations" even in this highly idealized setting indicates that surface temperature observations themselves may not be sufficient to constrain physically-important parameters such as

albedo, graybody thermal emissivity, or air-sea exchange coefficients separately. While state-space modeling approaches allow reconstruction of past surface climate states, it may be the case that the associated climate forcing may not contain sufficient information to extract the relative contributions of the individual physical processes that produced it. Further research will be necessary to understand whether the contribution of e.g. a single process like graybody thermal emissivity can be reliably estimated from the observations if regularized posteriors are used to constrain the other parameters of $g_\theta(u)$.

If the purpose of using the SEBM is to induce physical structure into the state reconstructions without specific concern regarding the parametric form of $g$, re-parametrization or nonparametric Bayesian inference can be used to estimate the form of the nonlinear function $g$ but avoid the ill-posedness of the parameter estimation problem. This is an option if the interest is in the posterior of the climate state and not in the individual contributions of energy sink and source processes.

State-of-the-art observation operators in paleoclimatology are often non-linear and contain non-Gaussian elements (Haslett et al., 2006; Tolwinski-Ward et al., 2011). A locally linearized observation model with data coming from the interpolation of proxy data can be used in the modeling framework we have considered, along with the assumption of Gaussian observation noise. Alternatively, it is also possible to first compute off-line point-wise reconstructions by inverting the full observation operator, potentially interpolating the results in time, and using a Gaussian approximation of the point-wise posterior distributions as observations in the SEBM (e.g. Parnell et al., 2016). We anticipate that such simplified observation operators will limit the accuracy of the parameter estimation, but that the regularized posterior would still be able to distinguish the most likely states and quantify the uncertainty in the estimation. Directly using non-linear, non-Gaussian observation operators requires a more sophisticated particle filter as optimal filtering is no longer possible. Such approaches will increase the computational cost and face difficulties avoiding filter degeneracy.

**The last paragraph in the conclusions is replaced by:**
This work shows that it is necessary to assess the well-posedness of the inverse problem of parameter estimation when reconstructing paleoclimate fields with physically motivated parametric stochastic models. In our case, the natural physical formulation of the SEBM is ill-posed. While climate states can be reconstructed, values of individual parameters are not strongly constrained by the observations. Regularized posteriors are a way to overcome the ill-posedness but retain a specific parametric form of the non-linear function representing the climate forcings.

**References**

Haslett, J., Whiley, M., Bhattacharya, S., Salter-Townshend, M., Wilson, S. P., Allen, J., Huntley, B., and Mitchell, F.: Bayesian paleoclimate reconstruction, Journal of the Royal Statistical Society, Series A, 169, 395–438, 2006.

Parnell, A. C., Haslett, J., Sweeney, J., Doan, T. K., Allen, J. R., and Huntley, B.: Joint Palaeoclimate reconstruction from pollen data via forward models and climate histories, Quarternary Science Reviews, 151, 111–126, 2016.

Tolwinski-Ward, S. E., Evans, M. N., Hughes, M. K., and Anchukaitis, K. J.: An efficient forward model of the climate controls on interannual variation in tree-ring width, Climate Dynamics, 36, 2419–2439, 2011.

---

## Author Comment (AC2) · 28 Jun 2019

Dear Referee,

We thank you for carefully reading the manuscript and for the valuable feedback. In the following, we respond to your comments and suggestions.

**Reviewer's comments:** Paleoclimate reconstruction seems to be an interesting and nonstandard question. So it might be worthy to give some details on how the SEBM is formulated, for example, what do each parameter $\theta$ stands for. It is also better to give

some references on why the specific parameters are chosen. For example, it seems that data are available with time-interval $\triangle = 0.01$ (what is the time unit here?). But is this practically true?

The SEBM follows the atmospheric model from Fanning and Weaver (1996), which contains parameterizations for incoming shortwave radiation, outgoing longwave radiation, radiative air-surface flux, sensible air-surface heat flux, and the latent heat flux into the atmosphere. The parametrizations contain terms of 0th, 1st, and 4th order, which are aggregated together such that roughly the 0th order terms correspond to incoming shortwave radiation and albedo effects, the 1st order terms correspond to air-sea heat (energy) exchange, and the 4th order terms correspond to longwave radiative transfer expressed as graybody emissivity. The prior distributions of $\theta$ aggregate the contributions of the different energy sources and sinks according to their parameterized polynomial order using the parameter and uncertainty estimates from Fanning and Weaver (1996). This first guess is adjusted using estimates of the current earth energy balance from Trenberth et al. (2009) to increase the physical consistency of the estimates. Thereby, at the equilibrium temperature, the contribution of the individual parameterized processes is very close to the estimates from Trenberth et al. (2009).

It should be noted that our parameterizations and the corresponding prior distributions are idealized. For example, spatial variations of the parameters are not yet included. However, we do not anticipate that increasing the realism of the SEBM would reduce the ill-posedness, as it would lead to a more complex parameter dependence structure without increasing the number of available (paleo-)observations.

We have briefly explained the prior range and model structure in the paragraphs in lines 28-30 on page 3 and 7-13 on page 4. We add the sentence "The nonlinear function $g_\theta(u)$ aggregates parametrizations from Fanning and Weaver (1996) for incoming shortwave radiation, outgoing long-wave radiation, radiative air-surface flux, sensible air-surface heat flux, and the latent heat flux into the atmosphere according to their polynomial order." to this description as additional clarification of the model structure.

The time unit is year. We add a sentence 'and one time unit represents a year" to line 6 in page 4. The time-interval $\triangle = 0.01$ represents about 4 days. In practice, missing or temporally integrated observations should be expected. Missing observations can be incorporated easily in our model, and would increase the reconstruction uncertainty similar to the reduction of the observed nodes (cf. Section 5.1). To include integrated observations, the observation operator can be interpolated (please see a new 'Section 5.4' in the revised manuscript in response to the first referee).

**Reviewer's comments:** While I agree the Fisher information matrix may be ill-conditioned, but I don't see immediately why the strong regularization approach is the right or natural way to fix it. The numerical results show that the regularized posterior has obvious biases, and sometimes close to being the prior. The strong regularization used here might be the cause of this. An alternative approach might be using the following version instead of (21)

$$p^N(\theta|u, y) \;\propto\; p(\theta)^\alpha [p_\theta(u)]^{1/N} \tag{1}$$

where $\alpha$ is a parameter in [0, 1], and it can be tuned for a better posterior.

We thank the reviewer for the above alternative approach, which is in line with our strongly regularized posterior (with $\alpha = 1$), and it aims to balance the contributions of the prior and the likelihood. As we admitted in the original manuscript, (page 12, line 16-17) the factor $1/N$ might not be optimal. However, it is not clear how to tune the optimal $\alpha$ or the factor $1/N$, and the development of a strategy for optimal regularization factors is beyond the scope of the current study. Therefore, we postpone it to future work. We have added a comment in the manuscript about this alternative approach. The need of a strongly regularized posterior is due to the degenerate likelihood, which is indicated by the ill-conditioned Fisher information matrix, as demonstrated numerically in Section 3.2 for different data sizes. In this case, the regularization by a prior in the standard Bayesian approach is not sufficient (as indicated by the Bernstein-von

Mises theorem in the asymptotic setting). Therefore, we need a stronger regularization that increases the contribution of the prior.

**Reviewer's comments:** Page 9, Fisher information matrix: since most NPG readers are likely to be geoscientists, maybe you should explain that in statistics, Fisher information dictates the asymptotic inference difficulty and give references. Also, there should be some explanations on why this matrix is ill-conditioned, not just some simulation plots.

While we agree with the reviewer that the Fisher information matrix dictates the asymptotic inference, we would prefer not to distract the readers to asymptotic inference, particularly because this manuscript focuses on non-asymptotic study.

Following the reviewer's suggestion, we added an intuitive argument on why the matrix is ill-conditioned on page 9 line 17: "As $N \rightarrow \infty$, the Fisher information matrix converges, by ergodicity of the system, to its expectation $\left( \Delta t \sigma_f^{-2} \mathbb{E}[(\mathbf{A} u_n)^{\circ k} A_{\mathcal{T}}^T \mathbf{C}^T \mathbf{C} A_{\mathcal{T}} (\mathbf{A} u_n)^{\circ l}] \right)_{k,l=0,1,4}$, where the matrices $\mathbf{A}$, $A_{\mathcal{T}}$ and $\mathbf{C}$, arising in the spatial-temporal discretization, are defined in Section A1. Intuitively, neglecting these matrices and viewing the vector $u_n$ as a scalar, this expectation matrix could be reduced to $(\Delta t \sigma_f^{-2} \mathbb{E}[u_n^k u_n^l])_{k,l=0,1,4}$, which is ill-conditioned because $u_n$ has a distribution concentrated near one with a standard deviation at the scale of $10^{-2}$ (see Figure 1)."

**Reviewer's comments:** Figure 2. The figure caption below "Data size $\log_{10} N$" is garbled. This happens to many figures later on as well. This might be a problem with my own computer/printer. But you better check.

We checked the captions, but garbled captions did not appear on our computers.

**Reviewer's comments:** Page 11, what is $u_i$? And it is better to define $u_c$ and $\sigma_o$ with

mathematical terms.

In Eq. (18), the subindex $i$ in $u_i$ indices the time. In the revised manuscript, $u_i$ is replaced by $u_n$ since $n$ is used throughout the manuscript. To avoid confusion of notation, we rewrote $u_c$ as $\overline{u}_c$. Since $u_c$ and $\sigma_o$ denote the mean and standard deviation of the observations, we believe that mathematical formulas for them are not necessary. In particular, they would lead to notational complexity.

**Reviewer's comments:** Page 13, I think Markov chain might be too abstract a term for NPG. I think you can replace it with MCMC for the same meaning.

We change the title of Sect. 4.1. to "Diagnosis of the Markov chain Monte Carlo algorithm". However, in the following MCMC refers to the method while Markov chain refers to the output of the MCMC algorithm. Thus, replacing Markov chain by MCMC throughout the section would be wrong and we still use Markov chain in the revised manuscript.

**Reviewer's comments:** Page 18, line 2, there should not be parenthesis for $\theta_0$, $\theta_1$.

We clarify the previous formulation by replacing it with "the correlations between $\theta_4$ and $\theta_0$ as well as $\theta_1$ are weakened".

**Reviewer's comments:** Page 29, line 7, "see e.g. [". You miss some content here.

Thanks for finding the typo. The correct form is "(see e.g. Doucet and Johansen, 2011)".

* * *
[Figure]

**Supplement:**

[revised manuscript text omitted]

---

## Referee Comment (RC3) · Anonymous Referee #2 · 1 Jul 2019

The authors have addressed all of my inquiries.